# Make Pre-trained Model Reversible:
# From Parameter to Memory Efficient Fine-Tuning

**Baohao Liao**     **Shaomu Tan**     **Christof Monz**
Language Technology Lab, University of Amsterdam
{b.liao, s.tan, c.monz}@uva.nl

## Abstract

Parameter-efficient fine-tuning (PEFT) of pre-trained language models (PLMs) has emerged as a highly successful approach, with training only a small number of parameters without sacrificing performance and becoming the de-facto learning paradigm with the increasing size of PLMs. However, existing PEFT methods are not memory-efficient, because they still require caching most of the intermediate activations for the gradient calculation, akin to fine-tuning. One effective way to reduce the activation memory is to apply a reversible model, so the intermediate activations are not necessary to be cached and can be recomputed. Nevertheless, modifying a PLM to its reversible variant is not straightforward, since the reversible model has a distinct architecture from the currently released PLMs. In this paper, we first investigate what is a key factor for the success of existing PEFT methods, and realize that it's essential to preserve the PLM's starting point when initializing a PEFT method. With this finding, we propose memory-efficient fine-tuning (MEFT) that inserts adapters into a PLM, preserving the PLM's starting point and making it reversible without additional pre-training. We evaluate MEFT on the GLUE benchmark and five question-answering tasks with various backbones, BERT, RoBERTa, BART and OPT. MEFT significantly reduces the activation memory up to 84% of full fine-tuning with a negligible amount of trainable parameters. Moreover, MEFT achieves the same score on GLUE and a comparable score on the question-answering tasks as full fine-tuning. A similar finding is also observed for the image classification task.[1]

## 1   Introduction

Large-scale pre-trained models have achieved great success across various domains and applications [1, 2, 3, 4, 5, 6, 7, 8]. As their capabilities continue to evolve, the released pre-trained language models (PLMs) have grown exponentially in size, even reaching a scale of 100 billion parameters [3, 9, 10, 11, 12]. Consequently, it presents unprecedented challenges in effectively leveraging these models for downstream tasks due to limited computing resources.

A historically common approach to adapting PLMs to downstream tasks is updating all pre-trained parameters, *full fine-tuning*. Although full fine-tuning has yielded numerous state-of-the-art results, its applicability is limited in storage-constrained environments. This constraint arises from maintaining a complete copy of the fine-tuned model for each task. An alternative

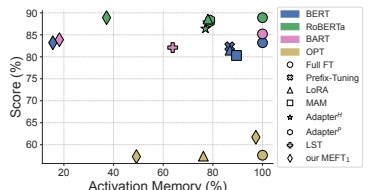

Figure 1: Average performance of different tasks vs. activation memory. The memory usage for full fine-tuning is denoted as 100%.

---

[1]Code at https://github.com/baohaoliao/mefts. Up-to-date version at https://arxiv.org/abs/2306.00477.

37th Conference on Neural Information Processing Systems (NeurIPS 2023).

adaptation approach is *parameter-efficient fine-tuning* (PEFT) [13, 14, 15, 16, 17, 18, 19] which involves selectively updating a small number of task-specific parameters while keeping the majority of the PLM's parameters frozen. PEFT offers significant advantages in reducing storage requirements by only saving one general PLM alongside the modified parameters for each task. In addition to storage savings, PEFT achieves comparable performance to full fine-tuning, sparking considerable interest in the adoption of PEFT.

Despite their advantages in parameter efficiency, existing PEFT methods still face challenges in terms of memory efficiency [20, 21]. PEFTs necessitate the caching of intermediate activations, similar to the requirements of full fine-tuning, to calculate the gradients of the trainable parameters. Typically, they consume more than 70% activation memory of full fine-tuning (see Figure 1). Since activations significantly contribute to the memory requirements during training, there are instances where fine-tuning a large-scale PLM with PEFT is not feasible due to memory constraints. To address this issue, a commonly employed approach is to treat the PLM as a feature extractor, such as knowledge distillation to a smaller model [22, 23], adding additional trainable layers on top [20] or aligned [21, 24] with it, and so on. These approaches circumvent the need to store the PLM's activations since the gradient computation graph does not traverse through the PLM. However, these methods often require additional pre-training or exhibit a substantial performance gap compared to full fine-tuning when using the same underlying model [20, 21].

In this paper, we propose a novel method called *memory-efficient fine-tuning* (MEFT) to modify PLMs in a parameter- and memory-efficient manner, without requiring additional pre-training. Initially, we investigate a crucial factor for the success of existing PEFT methods and determine that the proper initialization of newly added parameters is essential to maintain the continuity of information from the PLM (§2). Leveraging this insight, we design three MEFT methods that enable the modification of a PLM to its reversible variant, so it only necessitates caching the final output and allows for the recomputation of intermediate activations during back-propagation (§3). Consequently, MEFT significantly reduces the memory required for caching activations (see Figure 1).

To validate the effectiveness of our MEFT methods, we conduct extensive evaluations on the GLUE benchmark [25] with BERT [1], RoBERTa [2] and BART [26] (§4). The experimental results consistently demonstrate that our MEFT methods outperform both full fine-tuning and strong PEFT baselines in terms of parameter and memory efficiency. Remarkably, our methods achieve the same score as full fine-tuning while updating only 0.2% of the parameters and saving up to 84% of the activation memory. Furthermore, we evaluate MEFT on five question-answering tasks with a larger model, OPT [9]. The results show that our approach achieves a comparable score as full fine-tuning while saving 50% of the activation memory and updating only 0.64% of the parameters. A similar finding is also observed on the image classification task, SVHN [27]. Collectively, these experiments establish the effectiveness of MEFT as a powerful parameter- and memory-efficient approach that does not compromise performance.

## 2 Preliminaries

In this section, we aim to provide essential background knowledge by addressing the following questions: (1) Why are existing PEFTs not sufficiently memory-efficient (§2.1)? (2) What is a key factor for the success of PEFT (§2.2)? (3) What challenges does a reversible model have (§2.3)?

### 2.1 Parameter-efficient fine-tuning is not sufficiently memory-efficient

Given a $N$ multilayer perception: $\boldsymbol{h}_N = f_N(f_{N-1}(...(f_2(f_1(\boldsymbol{h}_0)))...))$ with $\boldsymbol{h}_0$ as the initial input, the $n^{th}$ layer $\boldsymbol{h}_n = f_n(\boldsymbol{h}_{n-1}) = \sigma_n(\boldsymbol{W}_n\boldsymbol{h}_{n-1})$ consists of a nonlinear function $\sigma_n$ and a weight matrix $\boldsymbol{W}_n$, where the bias term is ignored for simplicity. Denoting $\boldsymbol{x}_n = \boldsymbol{W}_n\boldsymbol{h}_{n-1}$, in backpropagation with a loss $\mathcal{L}$, the gradient of $\boldsymbol{W}_n$ is calculated with the chain rule as:

$$\frac{\partial \mathcal{L}}{\partial \boldsymbol{W}_n} = \frac{\partial \mathcal{L}}{\partial \boldsymbol{h}_N}\left(\prod_{i=n+1}^{N}\frac{\partial \boldsymbol{h}_i}{\partial \boldsymbol{x}_i}\frac{\partial \boldsymbol{x}_i}{\partial \boldsymbol{h}_{i-1}}\right)\frac{\partial \boldsymbol{h}_n}{\partial \boldsymbol{x}_n}\frac{\partial \boldsymbol{x}_n}{\partial \boldsymbol{W}_n} = \frac{\partial \mathcal{L}}{\partial \boldsymbol{h}_N}\left(\prod_{i=n+1}^{N}\boldsymbol{\sigma}_i'\boldsymbol{W}_i\right)\boldsymbol{\sigma}_n'\boldsymbol{h}_{n-1} \qquad (1)$$

where $\boldsymbol{\sigma}'$ is the derivative of $\sigma$ and the calculation of $\boldsymbol{\sigma}_n'$ requires $\boldsymbol{x}_n$. Therefore, $\{\boldsymbol{x}_i\}_{i=n}^{N}$ are cached during the forward pass to obtain the gradient of $\boldsymbol{W}_n$, even though $\{\boldsymbol{W}_i\}_{i>n}$ are frozen.

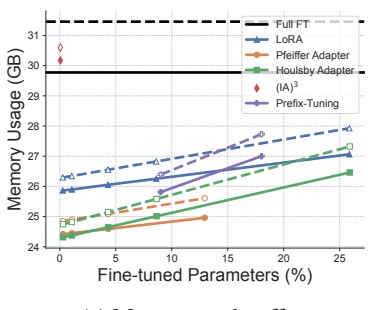
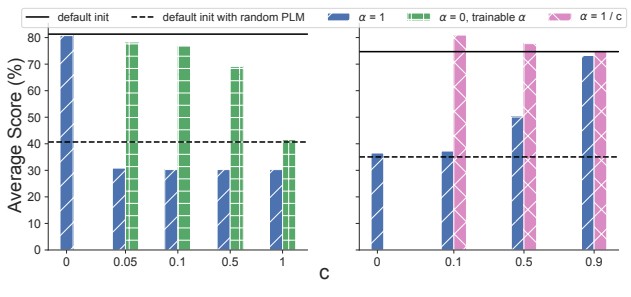

| (a) Memory trade-off. | (b) Initialization effect, Left: LoRA, Right: $(IA)^3$. |

Figure 2: Exploration of existing PEFTs: (a) The trade-off between memory and the number of trainable parameters. The dashed and solid lines denote the peak and activation memory, respectively. The model size for $BERT_{base}$ is $0.4GB^2$. (b) The initialization effect of PEFT on $RoBERTa_{base}$. Random PLM denotes that we initialize the backbone randomly instead of using a pre-trained model.

During training, the peak memory footprint is mainly occupied by three components: model's parameters $\{W_n\}_{n=1}^N$, optimizer state whose size is three times as large as the size of trainable parameters for Adam [28] (one for gradient and two for moments), and activations. The memory footprint for all three components is related to the model's depth and width. In addition, the memory footprint for activations is also related to some training settings, like batch size and sequence length.

Compared to full fine-tuning, existing PEFT methods, such as (Houlsby and Pfeiffer) Adapters [14, 16], LoRA [17], $(IA)^3$ [29], Prompt-Tuning [19] and Prefix-Tuning [15], tune a small number of parameters, making the size of the optimizer state negligible. However, the memory footprint required for activations is not significantly reduced. As shown in Figure 2a, where we set the batch size as 64 and the sequence length as 512 on RTE [30, 31, 32, 33] with $BERT_{base}$ [1], the activation memory of all PEFT methods is >75% of full fine-tuning, even with <1% trainable parameters.

## 2.2 Initialization is significant for parameter-efficient fine-Tuning

Pre-trained models learn generic and distributed enough representations to facilitate downstream learning of highly pressed task representation [36], i.e. offering a robust starting point for the training of downstream tasks. When modifying a PLM with PEFT, we hypothesize that one needs to preserve this starting point at the beginning of training for better performance.

**The Starting Point Hypothesis.** *When modifying a pre-trained model by adding new parameters, one needs to initialize the new parameters in a way to preserve the starting point from the pre-trained model at the beginning of training, such that fine-tuning the modified model can match the performance of full fine-tuning.*

More formally, supposed $f_n$ is a PLM layer and $h_n = f_n(h_{n-1})$, the output from a modified layer $f'_n$, $h'_n = f'_n(h_{n-1})$, should be close to $h_n$ at the beginning of training. I.e. $h'_n = h_n + \delta$, where $\|\delta\| \to 0$. Intuitively, we want $h'_n \approx h_n$, because $h'_n$ is the input to the next (modified) PLM layer. If they are dissimilar, the representation continuity will be broken down. Though most PEFT methods [14, 16, 17, 29] initialize their added modules in this way, we couldn't find a thorough investigation of the significance of this initialization in the existing literature. In this section, we explore the significance of PEFT's initialization for two methods, LoRA and $(IA)^3$ [29].

LoRA and $(IA)^3$ represent two common methods for introducing new parameters, involving addition and scaling operations, respectively. Given a pre-trained weight matrix $W \in \mathbb{R}^{d \times d}$, LoRA modifies it as $h' = (W + \frac{\alpha}{r} W_{down} W_{up})h$, where $W_{down} \in \mathbb{R}^{d \times r}$ and $W_{up} \in \mathbb{R}^{r \times d}$ are the added trainable parameters, $\alpha$ is a constant scale factor and normally $r \ll d$. LoRA's default initialization is $W_{down} \sim \mathcal{N}(0, \sigma^2)$ and $W_{up} = 0$. In this way, $W_{down} W_{up} = 0$ and the starting point from the

---

[2]Though we train in FP16, the PLM is first loaded in FP32, then auto-casted to FP16 for the forward pass in Transformers [34]. Since the memory required for the model in FP32 is always there during training, we report the memory for models in FP32 in this paper (see Table 9). More discussions about this are here. We believe it's a bug in the framework and can be resolved with further investigation. Especially Huggingface's new PEFT framework [35] allows loading INT8 model for fine-tuning.

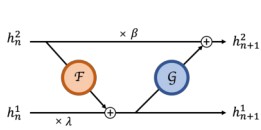

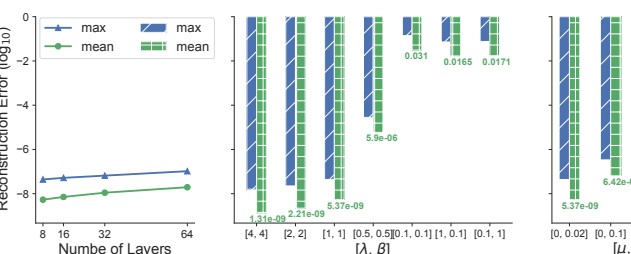

(a) Reversible architecture.

(b) Instability of reversible model.

Figure 3: (a) $\mathcal{F}$ and $\mathcal{G}$ are two arbitrary functions (sub-networks), taking two inputs, $\boldsymbol{h}_n^1$ and $\boldsymbol{h}_n^2$ (b) Reconstruction error between the vanilla and reversible gradients. The default setting is RevViT [40] with 8 layers, $\lambda = 1$, $\beta = 1$, $\mu = 0$ and $\sigma = 0.02$. Left: Different number of layers. Middle: Different scaling values. Right: Initialization with different means and standard deviations.

PLM is preserved perfectly. $(IA)^3$ modifies $\boldsymbol{W}$ by multiplying it to a trainable vector $\boldsymbol{l} \in \mathbb{R}^d$ as $\boldsymbol{h}' = (\boldsymbol{l} \odot \boldsymbol{W})\boldsymbol{h}$, where $\odot$ represents element-wise multiplication. The default initialization of $(IA)^3$ is $\boldsymbol{l} = \boldsymbol{1}$, also making the starting point untouched.

To facilitate the initialization process of LoRA, we opt for the following initial values: $\boldsymbol{W}_{down} = \boldsymbol{1}$, $\boldsymbol{W}_{up} = \boldsymbol{c}$ and $\alpha = 1$, where $\boldsymbol{c}$ is a matrix with all elements equal to an initialized value $c$, resulting in $\frac{\alpha}{r}\boldsymbol{W}_{down}\boldsymbol{W}_{up} = \boldsymbol{c}$. When $c = 0$, the starting point from a PLM is preserved. By adjusting $c$, we exert control over the degree of departure from the starting point. Similarly, we replace $\boldsymbol{l}$ with $\boldsymbol{l}' = \alpha\boldsymbol{l} = \alpha\boldsymbol{c}$ for $(IA)^3$.

In Figure 2b, we train the newly added parameters on RoBERTa[base] [2] for four tasks (CoLA [37], STS-B [38], MRPC [39] and RTE [30, 31, 32, 33]). For LoRA ($r = 8$), though we modify the initialization method, our result ($c = 0$) is very close to the default initialization. When the starting point is broken by $c \neq 0$ ($\alpha = 1$), all results are even worse than a randomly initialized model. However, when we set $\alpha = 0$[3] to preserve the starting point, all results become much better than the ones with $\alpha = 1$. For $(IA)^3$, when we decrease $c$ from 1 (default initialization) to 0, the results ($\alpha = 1$) become worse and worse. However, when we set $\alpha = 1/c$ to preserve the starting point, all results become better. Some of them are even better than the default initialization. All of the above-mentioned results show that it's significant to preserve the starting point from a PLM at the beginning of training when applying or designing a PEFT method. A different initialization scheme is in Figure 10 which leads to a similar finding.

## 2.3 Challenges of reversible neural network

Recapping a reversible model [41] in Figure 3a, one can reconstruct inputs from outputs as:

$$
\begin{aligned}
\boldsymbol{h}_{n+1}^1 &= \lambda\boldsymbol{h}_n^1 + \mathcal{F}_n(\boldsymbol{h}_n^2) & \boldsymbol{h}_n^2 &= (\boldsymbol{h}_{n+1}^2 - \mathcal{G}_n(\boldsymbol{h}_{n+1}^1))/\beta \\
\boldsymbol{h}_{n+1}^2 &= \beta\boldsymbol{h}_n^2 + \mathcal{G}_n(\boldsymbol{h}_{n+1}^1) & \boldsymbol{h}_n^1 &= (\boldsymbol{h}_{n+1}^1 - \mathcal{F}_n(\boldsymbol{h}_n^2))/\lambda
\end{aligned}
\tag{2}
$$

where $\lambda$ and $\beta$ are scaling factors. Theoretically, $\mathcal{F}_n$ and $\mathcal{G}_n$ could be two arbitrary functions (sub-networks). Given a multilayer reversible network, intermediate activations for each layer during the forward pass are not necessary to be cached. One only needs to store the final outputs, then reconstruct the intermediate activations and calculate the gradient layer-by-layer in a backward manner (See Listing 1 in §Appendix). In this way, the memory footprint required for activations can be reduced significantly and has no relationship with the model's depth, i.e. $\mathcal{O}(1)$ instead of $\mathcal{O}(N)$.

To investigate the training stability of a reversible model, we run experiments on RevViT [40].[4] RevViT shares the same architecture as Reformer [42], except applying a convolutional layer at the beginning to project an image into a sequence of vectors. When running RevViT, one could still cache the intermediate activations and treat it as an irreversible model. We term the gradient calculated in this way as *vanilla gradient*. One could also train RevViT in a reversible way, and the corresponding

---

[3]$\alpha$ has to be trainable when $\alpha = 0$. Otherwise, the newly added parameters are useless.

[4]Our experiments are based on this file, https://github.com/karttikeya/minREV/blob/main/rev.py.

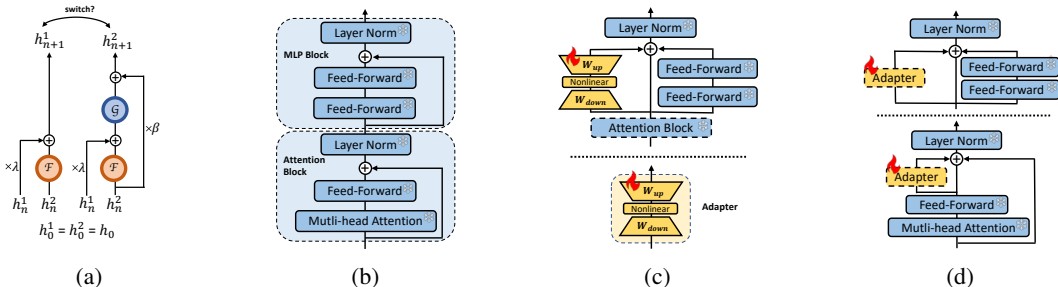

Figure 4: MEFT architectures. (a) Unfold reversible model. (b) A PLM layer. (c) Two MEFT architectures: (1) $\mathcal{F}$ is the PLM layer with an adapter (up) and $\mathcal{G}$ is an adapter (down); (2) $\mathcal{G}$ is the PLM layer with an adapter (up) and $\mathcal{F}$ is an adapter (down). (d) The third MEFT architecture: $\mathcal{G}$ is the MLP block with an adapter (up) and $\mathcal{F}$ is the attention block with an adapter (down). For initialization, $W_{down}, W_{up} \sim \mathcal{N}(0, \sigma^2)$ and $\sigma = 0.02$. Only the adapter is trainable.

gradient is called *reversible gradient*. We input the same random noises into the same RevViT twice to obtain the parameter gradients from the convolutional layer, in a vanilla and reversible way. Then we calculate the absolute difference between these two gradients and report the maximum and mean values. In this way, we want to check whether the vanilla gradient can be reconstructed in a reversible way. If the reconstruction error is large, it means that the vanilla gradient could not be recovered in a reversible way due to numerical stability, which might cause unstable training or bad performance.

As shown in Figure 3b, with an increasing number of layers in RevViT, the reconstruction error becomes larger, but still around $10^{-8}$ which is negligible. However, RevViT is sensitive to the scaling factors, $\lambda$ and $\beta$. When both scaling factors or one of them are less than $1$, the reconstruction error increases dramatically. We also explore the initialization of the linear layers in RevViT and find that a larger standard deviation or mean can cause a bigger reconstruction error. In sum, a larger number of layers, smaller scaling factors ($< 1$) and a larger standard deviation or mean for initialization tend to cause a bigger reconstruction error, which might result in the unstable training or low performance of a reversible model. Last but not least, RevViT [40] finds that residual connections inside $\mathcal{F}$ and $\mathcal{G}$ deteriorate the performance of a reversible Transformer [43].[5]

# 3  Memory-efficient fine-tuning

This paper aims to modify a PLM to its reversible variant without additional pre-training, so the PLM can still be fine-tuned with a limited memory footprint. The fundamental guiding principle behind our design is: preserving the starting point from a PLM to the greatest extent possible (discussion in §2.2). In this section, we propose three methods to modify a PLM to a reversible one.

Table 1: A summarization of three MEFT methods.

| MEFT? | $\mathcal{F}$ | $\mathcal{G}$ | $\lambda$ | $\beta$ | Switch | $h_n^1$ | $h_n^2$ |
|---|---|---|---|---|---|---|---|
| 1 | layer | adapter | $\to 0$ | any | ✓ | $\beta h_{n-1}$ | $h_n$ |
| 2 | adapter | layer | $\to 1$ | $\to 0$ | ✓ | $h_n$ | $h_{n-1}$ |
| 3 | attention | MLP | $\to 0$ | $\to 0$ | ✗ | $-$ | $h_n$ |

## 3.1  MEFT$_1$: PLM layer as $\mathcal{F}$, adapter as $\mathcal{G}$

As shown in Figure 4c, we design $\mathcal{F}$ as a pre-trained layer with an adapter, where the insertion position for the adapter is borrowed from He et al. [18]. $\mathcal{G}$ is simply an adapter. We initialize the adapters as $\boldsymbol{W}_{down}, \boldsymbol{W}_{up} \sim \mathcal{N}(0, \sigma^2)$, same for the following methods. In this way, the output from the adapter is close to $\boldsymbol{0}$ at the beginning of the training, so $\boldsymbol{h}_n \approx \mathcal{F}_n(\boldsymbol{h}_{n-1})$. For the following discussion, we only focus on the beginning of the training, making sure our design preserves the starting point from a PLM.

$\boldsymbol{h}_0$ and $\boldsymbol{h}_1$ are the input to and output from the $1^{st}$ layer of a PLM without any modification, respectively. I.e. $\boldsymbol{h}_0$ is the representation after the position and word embedding layers of a PLM. We

---

[5]In RevViT, $\mathcal{F}$ and $\mathcal{G}$ are the attention and MLP block (Figure 4b) without residual connection, respectively.

assign $\boldsymbol{h}_0^1 = \boldsymbol{h}_0^2 = \boldsymbol{h}_0$, same for the following methods. At the beginning of the training (see Figure 4a), $\boldsymbol{h}_1^1 = \lambda\boldsymbol{h}_0^1 + \mathcal{F}_1(\boldsymbol{h}_0^2) = \lambda\boldsymbol{h}_0 + \mathcal{F}_1(\boldsymbol{h}_0) \approx \lambda\boldsymbol{h}_0 + \boldsymbol{h}_1$, $\boldsymbol{h}_1^2 = \beta\boldsymbol{h}_0^2 + \mathcal{G}_1(\boldsymbol{h}_1^1) = \beta\boldsymbol{h}_0 + \mathcal{G}_1(\boldsymbol{h}_1^1) \approx \beta\boldsymbol{h}_0$, where the approximation holds because of our initialization of the adapters.

For now, $\boldsymbol{h}_1^1$ and $\boldsymbol{h}_1^2$ are not desired. When we input $\boldsymbol{h}_1^1$ and $\boldsymbol{h}_1^2$ to the $2^{nd}$ reversible layer, especially when we input $\boldsymbol{h}_1^2$ to $\mathcal{F}_2$, the representation continuity[6] is broken, because $\boldsymbol{h}_1^2 \neq \boldsymbol{h}_1$. We introduce two modifications to address this issue: (1) We set $\lambda \to 0$, so $\boldsymbol{h}_1^1 \approx \boldsymbol{h}_1$. (2) Then we switch the order of $\boldsymbol{h}_1^1$ and $\boldsymbol{h}_1^2$ before feeding to the next reversible layer, i.e. making $\boldsymbol{h}_1^1 \approx \beta\boldsymbol{h}_0$ and $\boldsymbol{h}_1^2 \approx \boldsymbol{h}_1$. In this way, $\boldsymbol{h}_1^2$ preserves the starting point. We don't require $\boldsymbol{h}_1^1$ to preserve any starting point, because it is entered to $\mathcal{G}_2$ which is not a pre-trained layer.

With the same above-mentioned design for the $2^{nd}$ reversible layer, we obtain $\boldsymbol{h}_2^1 \approx \beta\boldsymbol{h}_1$ and $\boldsymbol{h}_2^2 \approx \boldsymbol{h}_2$. By analogy, $\boldsymbol{h}_n^1 \approx \beta\boldsymbol{h}_{n-1}$ and $\boldsymbol{h}_n^2 \approx \boldsymbol{h}_n$, which means $\boldsymbol{h}_n^2$ always preserves the starting point from the PLM. Feeding $\boldsymbol{h}_n^2$ to the next reversible layer, $\mathcal{F}_{n+1}$, doesn't break the representation continuity. After all layers, we input $\boldsymbol{h}_N' = (\boldsymbol{h}_N^1 + \boldsymbol{h}_N^2)/2$ to a task-specific head that is a brand new layer, same for the following methods.[7]

## 3.2 MEFT$_2$: Adapter as $\mathcal{F}$, PLM layer as $\mathcal{G}$

Opposite to MEFT$_1$, we design $\mathcal{F}$ as an adapter and $\mathcal{G}$ as the PLM layer with an adapter for MEFT$_2$ (see Figure 4c). In this case, we need to make sure that the input to $\mathcal{G}$ preserves the starting point. Let's also start with the first layer, $\boldsymbol{h}_1^1 = \lambda\boldsymbol{h}_0^1 + \mathcal{F}_1(\boldsymbol{h}_0^2) = \lambda\boldsymbol{h}_0 + \mathcal{F}_1(\boldsymbol{h}_0) \approx \lambda\boldsymbol{h}_0$, $\boldsymbol{h}_1^2 = \beta\boldsymbol{h}_0^2 + \mathcal{G}_1(\boldsymbol{h}_1^1) = \beta\boldsymbol{h}_0 + \mathcal{G}_1(\boldsymbol{h}_1^1) \approx \beta\boldsymbol{h}_0 + \mathcal{G}_1(\lambda\boldsymbol{h}_0)$, where the approximation holds because of our initialization of the adapters.

To preserve the starting point from the PLM, we set $\lambda \to 1$, $\beta \to 0$ and switch the order of $\boldsymbol{h}_1^1$ and $\boldsymbol{h}_1^2$ before feeding to the next reversible layer. When setting $\lambda \to 1$, we make sure the representation continuity is preserved for $\mathcal{G}_1$, resulting in $\boldsymbol{h}_1^2 \approx \beta\boldsymbol{h}_0 + \boldsymbol{h}_1$. When $\beta \to 0$ and the order of $\boldsymbol{h}_1^1$ and $\boldsymbol{h}_1^2$ is switched, $\boldsymbol{h}_1^1 \approx \boldsymbol{h}_1$ and $\boldsymbol{h}_1^2 \approx \boldsymbol{h}_0$. In this way, $\boldsymbol{h}_1^1$ preserves the initialization point, and we won't break the representation continuity when feeding it to $\mathcal{G}_2$ in the next reversible layer. With the same setting for each layer, $\boldsymbol{h}_n^1 \approx \boldsymbol{h}_n$ and $\boldsymbol{h}_n^2 \approx \boldsymbol{h}_{n-1}$, so $\boldsymbol{h}_n^1$ always preserves the starting point.

## 3.3 MEFT$_3$: Attention block as $\mathcal{F}$, MLP block as $\mathcal{G}$

As shown in Figure 4d, we can also design $\mathcal{F}$ as the pre-trained attention block with an adapter and $\mathcal{G}$ as the pre-trained MLP block with an adapter. Also starting with the first layer, we obtain $\boldsymbol{h}_1^1 = \lambda\boldsymbol{h}_0^1 + \mathcal{F}_1(\boldsymbol{h}_0^2) = \lambda\boldsymbol{h}_0 + \mathcal{F}_1(\boldsymbol{h}_0)$, $\boldsymbol{h}_1^2 = \beta\boldsymbol{h}_0^2 + \mathcal{G}_1(\boldsymbol{h}_1^1) = \beta\boldsymbol{h}_0 + \mathcal{G}_1(\boldsymbol{h}_1^1)$.

$\lambda \to 0$ is required, so $\boldsymbol{h}_1^1$ approximates the original output from the pre-trained attention block, and can be fed to $\mathcal{G}_1$ to preserve the starting point. $\beta \to 0$ is also required, so $\boldsymbol{h}_1^2 \approx \boldsymbol{h}_1$, and can be fed to $\mathcal{F}_2$ in the next reversible layer. By default, we set $\lambda = \beta \to 0$. For MEFT$_3$, one doesn't need to switch the order of $\boldsymbol{h}_1^1$ and $\boldsymbol{h}_1^2$ before feeding to the next reversible layer. For each layer, $\boldsymbol{h}_n^1$ is close to the original output from the attention block of the corresponding PLM layer, and $\boldsymbol{h}_n^2 \approx \boldsymbol{h}_n$.

Compared to the vanilla RevNet [41] where $\lambda = \beta = 1$, we meticulously assign different values to $\lambda$ and $\beta$ to preserve the starting point from a PLM, and switch the order of the outputs before feeding to the next layer (if necessary) to preserve the representation continuity. We summarize the settings for all three MEFT methods in Table 1.

# 4 Experiments

## 4.1 Experimental setup

**Datasets and evaluation.** We evaluate MEFTs on eight sequence representation tasks and five sequence-to-sequence tasks. All sequence representation tasks are from the GLUE benckmark [25].

---

[6]The presentation continuity and the starting point hypothesis emphasize two aspects. The presentation continuity, for example, shows that one can't feed $\boldsymbol{h}_0$ to the third pre-trained layer, focusing on the input. The starting point hypothesis shows that the output from a modified pre-trained layer should be close to the output from the original pre-trained layer, focusing on the output. However, they are also very related, since the output from the current layer is the input to the next layer.

[7]Read Appendix §C for a step-by-step tutorial if you still feel confused.

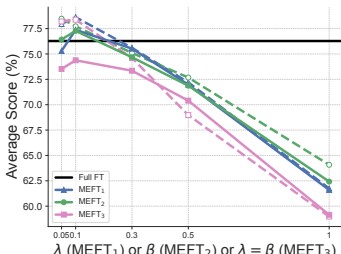
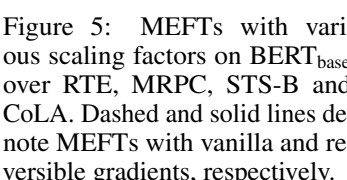
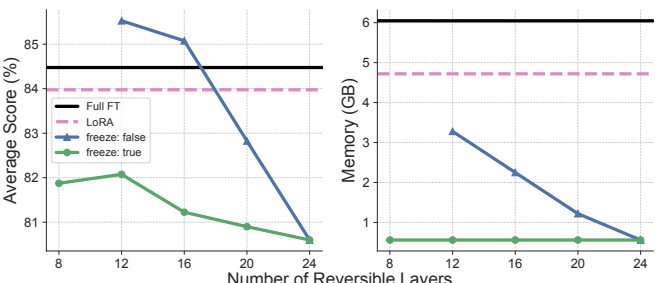

Figure 5: MEFTs with various scaling factors on BERT$_{base}$ over RTE, MRPC, STS-B and CoLA. Dashed and solid lines denote MEFTs with vanilla and reversible gradients, respectively.

Figure 6: The trade-off between the performance and activation memory with MEFT$_1$ on RoBERTa$_{large}$ over RTE, MRPC, STS-B and CoLA. The line annotated by 'freeze: true' means the shallower PLM layers are frozen without any adaptation, while the line annotated by 'freeze: false' means the top MEFT layers with vanilla gradient, as shown in Figure 7.

The sequence-to-sequence tasks are question-answering benchmarks, including OpenBookQA [44], PIQA [45], ARC (easy and challenge) [46] and SciQ [47]. We show the statistics of these datasets in Table 8 in Appendix. For the GLUE benchmark, we report accuracy on MNLI, QQP, QNLI, SST-2, MRPC and RTE, Pearson correlation coefficient on STS-B (if not specially mentioning) and Matthews correlation coefficient [48] on CoLA. We report accuracy on all question-answering tasks. In addition, we report all results on the development sets as our baselines.

**Models.** We use the encoder-only models (BERT$_{base}$ [1], RoBERTa$_{large}$ [2] and BART$_{large}$ encoder [26]) as the underlying models for all GLUE tasks, and the decoder-only models (OPT$_{1.3B}$ and OPT$_{6.7B}$ [9]) for question-answering tasks. (See Table 9 in Appendix for model details.)

**Baselines.** The most important baseline is full fine-tuning (**Full FT**) that updates all parameters. Houlsby Adapter (**Adapter$^H$**) [14], Pfeiffer Adapter (**Adapter$^P$**) [16], **Prefix-Tuning** [15] and **LoRA** [17] are chosen as PEFT baselines. In addition, two unified PEFT methods, **MAM** [18] and **AutoPEFT** [49], that combine multiple PEFT methods are also chosen as PEFT baselines. Lastly, two feature-based tuning methods, $\mathcal{Y}$-**Tuning** [20] and **LST**[21], that aim to reduce training memory serve as memory-efficient baselines. We report the baseline results from the original papers if possible.

**Implementation.** For computational efficiency, we set $\beta = 1$ for MEFT$_1$, $\lambda = 1$ for MEFT$_2$, and only tune the factors that are required $\rightarrow 0$ (see Table 1). After obtaining the optimal value, i.e. 0.1, we use this value for all three MEFT methods, tasks and backbones. On the GLUE benchmark, we sweep learning rates in $\{3, 4, 5\}\cdot10^{-4}$, batch sizes in $\{16, 32\}$ and the number of epochs in $\{10, 20\}$ for the tasks with >10k training samples. For the low-resource tasks with <10k training samples, we sweep learning rates in $\{5, 6, 7, 8\}\cdot10^{-4}$, batch sizes in $\{16, 32\}$ and the number of epochs in $\{20, 40\}$. These grid search spaces are inspired by our baselines, especially by LoRA [17]. We use the default Adam [28] setting with a warmup ratio of 6%. If the model's performance on the development set is not improved over 5 epochs, we stop the training. We run the same task of a method in the above-mentioned grid search space three times with different random seeds, choose the best result from each run, and report the mean and standard deviation of these best results. For all question-answering tasks, we sweep learning rates in $\{1, 3, 5, 7\}\cdot10^{-4}$, batch sizes in $\{8, 16, 32\}$ and the number of epochs in $\{3, 5, 10\}$, and keep other settings the same, which is inspired by [50]. The sequence length for all tasks is set to 512, 128, 128 and 128 for BERT$_{base}$, RoBERTa$_{large}$, BART$_{large}$ and OPT as our baselines, respectively. We run all experiments on the Transformers framework [34] on a single NVIDIA RTX A6000 GPU with 48GB memory. Overall, a single run of any task could be finished within 8 hours, and most tasks could be finished in an hour. Fine-tuning settings are summarized in Table 7.

## 4.2 Results and discussions

**Importance of MEFT's initialization.** In the beginning, we further test the starting point hypothesis on our MEFTs by adjusting the scaling factors, $\lambda$ and $\beta$. As depicted by the dashed lines in Figure 5, the degradation in performance is evident when the scaling factors deviate from their desired value of

Table 2: Comparion with different methods on GLUE. The first and second best results are in **bold** and underlined, respectively. All baseline results for BERT$_{base}$ and RoBERTa$_{large}$ are from [49] and [17], respectively. We report Spearman's Correlation for STS-B and matched accuracy for MNLI on BERT$_{base}$. The hyper-parameters after each backbone are used for measuring the memory footprint. $r = 8$ for all MEFTs. All models are trained in FP16 if not specified with FP32.

| Method | #Param. (%) | Memory (GB) Peak | Act. | RTE | MRPC | STS-B | CoLA | SST-2 | QNLI | QQP | MNLI | Avg. |
|---|---|---|---|---|---|---|---|---|---|---|---|---|
| *BERT$_{base}$* (batch size = 32, sequence length = 512) | | | | | | | | | | | | |
| Full FT | 100 | 16.67 | 14.98 | $71.1_{1.5}$ | $85.7_{1.8}$ | $89.0_{0.5}$ | $59.3_{0.6}$ | $92.6_{0.2}$ | $91.5_{0.1}$ | **$91.5_{0.0}$** | $84.4_{0.2}$ | 83.2 |
| Prefix-Tuning | 0.17 | 13.58 | 13.00 | $70.5_{0.5}$ | $85.9_{0.9}$ | $88.8_{0.2}$ | $58.9_{1.2}$ | $91.9_{0.5}$ | $90.8_{0.1}$ | $89.1_{0.1}$ | $82.8_{0.2}$ | 82.3 |
| LoRA | 0.27 | 13.45 | 13.02 | $65.9_{1.5}$ | $84.5_{1.0}$ | $88.7_{0.1}$ | $57.6_{0.8}$ | $92.1_{0.4}$ | $90.6_{0.2}$ | $89.4_{0.0}$ | $83.0_{0.1}$ | 81.5 |
| MAM | 6.97 | 14.21 | 13.41 | $69.1_{1.8}$ | $87.2_{0.7}$ | $89.0_{0.5}$ | $47.9_{24.}$ | $83.9_{17.}$ | $90.9_{0.2}$ | $90.8_{0.1}$ | $83.3_{0.2}$ | 80.3 |
| AutoPEFT | 1.40 | - | - | $72.4_{0.9}$ | **$87.5_{0.9}$** | $89.2_{0.0}$ | $60.9_{1.5}$ | $92.1_{0.3}$ | $91.1_{0.1}$ | $90.6_{0.1}$ | $84.0_{0.1}$ | 83.5 |
| *vanilla gradient* | | | | | | | | | | | | |
| MEFT$_1$ | 0.27 | 13.64 | 13.21 | $74.2_{1.4}$ | $86.7_{0.2}$ | $89.0_{0.0}$ | $62.1_{0.2}$ | $92.9_{0.2}$ | **$91.6_{0.1}$** | $89.9_{0.1}$ | $83.8_{0.4}$ | 83.8 |
| MEFT$_2$ | 0.27 | 13.73 | 13.31 | $74.7_{0.3}$ | $86.6_{0.5}$ | **$89.4_{0.1}$** | $61.8_{0.7}$ | **$93.0_{0.1}$** | **$91.6_{0.1}$** | $90.2_{0.1}$ | **$84.5_{0.1}$** | 84.0 |
| MEFT$_3$ | 0.27 | 13.64 | 13.21 | **$76.1_{0.8}$** | $87.4_{0.3}$ | $88.9_{0.1}$ | **$62.3_{0.5}$** | **$93.2_{0.2}$** | $91.5_{0.1}$ | $90.1_{0.1}$ | $84.2_{0.2}$ | **84.2** |
| *reversible gradient* | | | | | | | | | | | | |
| MEFT$_1$(FP32) | 0.27 | 2.75 | 2.33 | $73.9_{0.5}$ | $86.5_{0.2}$ | $88.8_{0.1}$ | $60.3_{0.6}$ | $92.7_{0.4}$ | $91.4_{0.0}$ | $88.8_{0.1}$ | $83.4_{0.1}$ | 83.2 |
| MEFT$_2$(FP32) | 0.27 | 3.53 | 3.11 | $74.0_{0.6}$ | $86.3_{0.4}$ | $88.6_{0.1}$ | $60.7_{1.5}$ | $92.8_{0.2}$ | $91.5_{0.1}$ | $88.9_{0.1}$ | $83.1_{0.1}$ | 83.2 |
| MEFT$_3$(FP32) | 0.27 | 2.99 | 2.57 | $70.8_{0.6}$ | $84.6_{0.5}$ | $88.2_{0.3}$ | $53.9_{1.0}$ | $92.2_{0.4}$ | $90.4_{0.2}$ | $86.9_{0.3}$ | $81.5_{0.1}$ | 81.1 |
| *RoBERTa$_{large}$* (batch size = 32, sequence length = 128) | | | | | | | | | | | | |
| Full FT | 100 | 11.47 | 6.05 | 86.6 | 90.9 | **92.4** | 68.0 | 96.4 | 94.7 | **92.2** | 90.2 | 88.9 |
| Adapter$^H$ | 0.23 | 6.05 | 4.66 | $72.9_{3.0}$ | $87.7_{1.7}$ | $91.5_{0.5}$ | $66.3_{2.0}$ | $96.3_{0.5}$ | $94.7_{0.2}$ | $91.5_{0.1}$ | $90.3_{0.3}$ | 86.4 |
| Adapter$^H$ | 1.69 | 6.18 | 4.71 | $83.4_{1.1}$ | $88.7_{2.9}$ | $91.0_{1.7}$ | $66.5_{4.4}$ | $96.2_{0.3}$ | $94.7_{0.2}$ | $92.1_{0.1}$ | $89.9_{0.5}$ | 87.8 |
| Adapter$^P$ | 0.23 | 6.16 | 4.77 | $80.1_{2.9}$ | $89.7_{1.2}$ | $91.9_{0.4}$ | $67.8_{2.5}$ | $96.6_{0.2}$ | $94.8_{0.3}$ | $91.7_{0.2}$ | $90.5_{0.3}$ | 87.9 |
| Adapter$^P$ | 0.85 | 6.21 | 4.78 | $83.8_{2.9}$ | $90.2_{0.7}$ | $92.1_{0.7}$ | $68.3_{1.0}$ | $96.1_{0.3}$ | $94.8_{0.2}$ | $91.9_{0.1}$ | $90.2_{0.3}$ | 88.4 |
| LoRA | 0.23 | 6.11 | 4.72 | $85.2_{1.1}$ | $90.2_{1.0}$ | $92.3_{0.5}$ | $68.2_{1.9}$ | $96.2_{0.5}$ | $94.8_{0.3}$ | $91.6_{0.2}$ | **$90.6_{0.2}$** | 88.6 |
| *vanilla gradient* | | | | | | | | | | | | |
| MEFT$_1$ | 0.23 | 6.19 | 4.81 | $89.5_{0.8}$ | **$91.5_{0.2}$** | $92.3_{0.1}$ | **$69.9_{0.7}$** | **$96.8_{0.1}$** | **$94.9_{0.1}$** | $91.5_{0.1}$ | $90.3_{0.2}$ | **89.6** |
| MEFT$_2$ | 0.23 | 6.20 | 4.82 | $88.6_{0.6}$ | $91.3_{0.4}$ | $92.2_{0.1}$ | $68.8_{0.7}$ | **$96.8_{0.1}$** | $94.8_{0.1}$ | $91.4_{0.1}$ | **$90.6_{0.0}$** | 89.3 |
| *reversible gradient* | | | | | | | | | | | | |
| MEFT$_1$ | 0.23 | 3.11 | 1.73 | $87.6_{0.3}$ | $90.5_{0.6}$ | $91.6_{0.1}$ | $63.3_{1.7}$ | $95.9_{0.1}$ | $94.3_{0.2}$ | $90.1_{0.1}$ | $89.2_{0.7}$ | 87.8 |
| MEFT$_1$(FP32) | 0.23 | 3.63 | 2.25 | **$90.0_{0.5}$** | $91.2_{0.2}$ | **$92.4_{0.1}$** | $66.1_{0.7}$ | $96.7_{0.3}$ | $94.8_{0.1}$ | $90.2_{0.5}$ | $90.1_{0.1}$ | 88.9 |
| MEFT$_2$(FP32) | 0.23 | 3.75 | 2.37 | $88.2_{0.5}$ | $90.5_{0.4}$ | $92.1_{0.0}$ | $64.4_{0.6}$ | $95.9_{0.2}$ | $94.3_{0.1}$ | $89.4_{0.1}$ | $88.4_{0.5}$ | 87.9 |
| *BART$_{large}$* (batch size = 100, sequence length = 128) | | | | | | | | | | | | |
| Full FT [20] | 100 | 12.75 | 9.62 | 77.6 | **89.2** | - | 59.3 | 95.8 | 94.3 | **89.5** | 90.8 | 85.2 |
| $\mathcal{Y}$-Tuning [20] | 7.7 | - | - | 62.8 | 79.2 | - | 44.4 | 94.4 | 88.2 | 85.5 | 82.3 | 76.7 |
| LST(FP32) [21] | 2.6 | 7.05 | 6.14 | 69.7 | 87.3 | - | 55.5 | 94.7 | 91.9 | **89.5** | 86.1 | 82.1 |
| *reversible gradient* | | | | | | | | | | | | |
| MEFT$_1$ | 0.20 | 2.54 | 1.75 | $72.2_{1.3}$ | $88.1_{1.3}$ | - | $51.0_{1.8}$ | $95.1_{0.2}$ | $92.4_{0.1}$ | $87.5_{0.0}$ | $87.0_{0.2}$ | 81.9 |
| MEFT$_1$(FP32) | 0.20 | 2.54 | 1.75 | $74.3_{0.7}$ | $88.4_{0.5}$ | - | $57.4_{2.2}$ | $95.4_{0.1}$ | $93.9_{0.1}$ | $89.3_{0.1}$ | $88.3_{0.1}$ | 83.9 |

0 (as indicated in Table 1). However, when they are small enough (0.05 or 0.1), the results are even better than full fine-tuning. For most MEFT methods (MEFT$_1$ and MEFT$_3$), the optimal value for the scaling factors is 0.1. So we use this value for all MEFT methods in the following experiments.

**MEFTs with vanilla gradient are strong PEFT methods.** Though MEFTs have reversible architectures, we can still treat them as irreversible models and cache the intermediate activations during fine-tuning. In this way, they are simply PEFT methods. In Table 2, all MEFT methods, utilizing the vanilla gradient, consistently outperform both full fine-tuning and other baseline approaches by a significant margin. For example, MEFT$_3$ outperforms Full FT by 1% and the best PEFT baseline (AutoPEFT) by 0.7% on BERT$_{base}$. MEFT$_1$ outperforms Full FT by 0.7% on RoBERTa$_{large}$.

**Performance gap of MEFTs between vanilla and reversible gradients.** In Figure 5, the results of MEFTs with reversible gradient (solid line) are often lower than the ones with vanilla gradient (dashed line). Recapping the discussion in §2.3, smaller scaling factors ($< 1$) and residual connections in $\mathcal{F}$ and $\mathcal{G}$ can cause a larger reconstruction error because of numerical stability. When modifying a PLM, we can't remove the residual connections from it and have to set the scaling factors $\to 0$ due to the starting point hypothesis, which we believe is the main reason for the performance drop. Our claim is further supported by MEFT$_3$ which has the most evident drop among all MEFTs. Compared to MEFT$_1$ and MEFT$_2$ that only have a residual connection in either $\mathcal{F}$ or $\mathcal{G}$, both $\mathcal{F}$ and $\mathcal{G}$ of MEFT$_3$ have residual connections. In addition, we have to set both $\lambda$ and $\beta$ close to 0 for MEFT$_3$, which also causes a bigger reconstruction error than only setting one scaling factor (see Figure 3b middle). Since MEFT$_3$ with reversible gradient performs the worst among all MEFTs, we only run it on BERT$_{base}$

due to limited resources. Expectedly, $MEFT_1$ trained in FP32 outperforms it trained in FP16 on both RoBERTa$_{large}$ and BART$_{large}$ (see Table 2), because FP16 causes more instability.

**Reversible MEFTs on deep model.** Because of the starting point hypothesis, the residual connection from PLMs remains and the scaling factors are set closely to 0. With an increasing number of layers, the training instability is expected to become more severe (see Figure 3b left). As shown in Figure 6, when all RoBERTa layers are reversible (the number of reversible layers as 24), the score drops dramatically. To address this issue, we propose three settings in Figure 7: (1) Cache the activations for top layers (vanilla gradient) and apply reversible shallow layers (reversible gradient). (2) Freeze some shallow PLM layers, i.e. treating the shallow layers as a feature extractor. (3) Combine the above two settings. Notably, we have to put the reversible layers under the vanilla layers due to numerical stability. If we reverse the order, the reconstruction error is transferred to the vanilla layers.

We only explore the first two settings on RoBERTa and will discuss the third setting in the following, since RoBERTa$_{large}$ doesn't contain many layers. In Figure 6, when we apply the first setting (freeze: false) to RoBERTa$_{large}$, the average score becomes better when the number of reversible layers decreases, outperforms full fine-tuning when it's $\leq 16$. However, the activation memory also increases with an increasing number of vanilla layers, since the vanilla layers require caching the activations. By default, we set the number of reversible layers as 16 for RoBERTa$_{large}$ in Table 2. For the second setting (freeze: true), the results are always worse than full fine-tuning. However, its activation memory stays the same since all trainable layers are reversible.

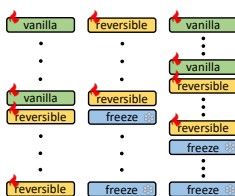

Figure 7: Three settings for deep models.

**MEFTs are parameter and memory-efficient with a strong performance.** Let's go back to Table 2. Though there is a gap in MEFTs between vanilla and reversible gradients, reversible MEFTs still achieve strong results compared to previous baselines. On BERT$_{base}$, reversible $MEFT_1$ and $MEFT_2$ obtain the same average score as Full FT, slightly worse than the best PEFT method, AutoPEFT (83.2 vs. 83.5). However, reversible MEFTs only requires about 21% and 24% activation memory of Full FT and PEFTs. On RoBERTa$_{large}$, reversible $MEFT_1$ (FP32) achieves the same score as Full FT and outperforms all PEFT methods, while only requiring 37% and 48% activation memory of Full FT and PEFTs.

Due to limited computing resources, we only conduct experiments on the best MEFT method, $MEFT_1$, on BART$_{large}$ when compared to other memory-efficient methods. In addition, we don't use our own grid search space on BART$_{large}$. Instead, we apply the same grid search space as LST, setting the learning rate in $\{3 \cdot 10^{-4}, 1 \cdot 10^{-3}, 3 \cdot 10^{-3}\}$, the batch size as 100 and the number of epochs as 20. In this way, we want to validate the robustness of MEFT. Similarly, $MEFT_1$ outperforms the memory-efficient baselines by a large margin while only requiring 29% LST's activation memory. In addition, LST requires knowledge distillation to initialize the added layers and is not stable [21].[8]

**MEFT trained in FP32 vs. in FP16, and the time-memory tradeoff.** Though reversible MEFTs trained in FP32 outperform the ones trained in FP16, there are still some notable discussions about them: (1) The memory footprint required by reversible MEFTs trained in FP32 and FP16 is the same. In Table 2, $MEFT_1$ and $MEFT_1$ (FP32) have the same activation memory on BART$_{large}$, because the recomputed activations in back-propagation are always in FP32 due to the mixed precision training [51]. I.e. PyTorch [52] only allows FP32 in back-propagation; (2) FP16 still benefits the training of large PLMs. In Table 2, the peak and activation memory difference is about the backbone size in FP32 for PEFTs and MEFTs. If one could reduce the backbone size by loading in FP16, we can further reduce the peak memory;[2] (3) Training in FP16 is faster than the training in FP32 (about 1:1.2) due to the forward pass. In addition, since reversible MEFTs recompute the activations, they require more training time, about twice the training time for MEFTs with the vanilla gradient.

**Results on larger and deeper models.** Here we explore a more realistic setting (the third setting in Figure 7) on larger and deeper models, OPT$_{1.3B}$ and OPT$_{6.7B}$, in Table 3. On OPT$_{1.3B}$ with 24 layers, we set the number of frozen, reversible and vanilla layers as 8. On OPT$_{6.7B}$ with 32 layers, we use 8 reversible and vanilla layers, same as OPT$_{1.3B}$. For a fair comparison, we freeze the first 8 PLM layers and modify the rest 16 layers with LoRA. $MEFT_1$ is comparable to LoRA, while only requiring LoRA's 65% activation memory. Though slightly worse than Full FT (-0.3%), $MEFT_1$'s

---

[8]The comparison of memory footprint to $\mathcal{Y}$-Tuning is in Table 10.

Table 3: Results on question-answering tasks. $r = 64$ for both $\text{MEFT}_1$ and LoRA. All methods are trained in FP16. Due to limited computing resources, we only conduct one random run with these methods. A batch size of 32 and a sequence length of 128 are used to measure the memory footprint and training time. The training time is for one epoch on the OpenBookQA task. Check Appendix §D for the implementation detail.

| Model | Method | #Param. (%) | Memory (GB) Peak | Memory (GB) Activation | Time (s) | OpenBookQA | PIQA | ARC-E | ARC-C | SciQ | Avg. |
|---|---|---|---|---|---|---|---|---|---|---|---|
| $\text{OPT}_{1.3B}$ | Full FT [50] | 100 | 28.31 | 8.23 | 128.0 | 31.4 | 75.2 | 61.3 | 27.7 | 92.5 | 57.6 |
| | LoRA | 0.64 | 11.42 | 6.27 | 36.6 | 29.9 | 74.9 | 60.1 | 28.7 | 93.3 | 57.4 |
| | $\text{MEFT}_1$ | 0.64 | 9.20 | 4.05 | 45.2 | 34.0 | 73.1 | 57.1 | 28.8 | 93.1 | 57.3 |
| $\text{OPT}_{6.7B}$ | ZeroShot | - | - | - | - | 27.6 | 76.2 | 65.6 | 30.6 | 90.1 | 58.0 |
| | $\text{MEFT}_1$ | 0.25 | 33.67 | 8.01 | 200.4 | 37.0 | 77.4 | 65.7 | 34.1 | 94.4 | 61.7 |

activation memory is only half of the one for Full FT. When using the same activation memory as Full FT by running on $\text{OPT}_{6.7B}$, $\text{MEFT}_1$ outperforms Full FT by a large margin.

**Transfer to image classification task.** Though we only focused on NLP tasks, MEFT could be transferred to other tasks, even other architectures. We leave the transfer of MEFT to other architectures for future work, and here apply MEFT to ViT [53] for an image classification task, i.e. SVHN [27]. We follow the main training recipe from AdaptFormer [54], except for changing the optimizer from SGD to AdamW, setting the maximum gradient norm as

Table 4: Results on image classification.

| Method | Acc@1 | Peak Memory (GB) |
|---|---|---|
| Full FT [27] | 97.67 | - |
| AdaptFormer [27] | 96.89 | 36 |
| $\text{MEFT}_1$ | 96.74 | 9 |

0.3. For $\text{MEFT}_1$'s hyper-parameters, we set $r = 64$ and $\lambda = 0.3$ (smaller $\lambda$ is not stable for training). Similar to the NLP's results, $\text{MEFT}_1$ achieves comparable accuracy as AdaptFormer while saving a large amount of memory footprint in Table 4.

For more results about comparing MEFT to gradient checkpointing, comparing MEFT to quantization methods, and combining MEFT with other memory-efficient methods, please go to Appendix §E. In addition, due to the page limit, we put the detailed related works in Appendix §A, and discuss the limitation of our work in Appendix §B.

## 5 Conclusion

In this paper, we propose three memory-efficient fine-tuning methods (MEFTs), that fine-tune PLM in a parameter-efficient and memory-efficient way without the requirement of additional pre-training and match the performance of full fine-tuning. MEFTs modify the PLM architecture with adapters and make it reversible, by following the starting point hypothesis that is essential for PEFTs. So MEFTs don't require caching the intermediate activations during training and significantly reduce the memory footprint occupied by activations. When applying MEFTs to various models, BERT, RoBERTa and BART, on the GLUE benchmark, MEFTs achieve a similar score as full fine-tuning and other strong baselines, while saving up to 84% activation memory. A similar story is also observed when applying MEFT to larger and deeper models, OPT, on five question-answering tasks. MEFT achieves a comparable score as full fine-tuning and only consumes its 50% activation memory. However, because of the recomputation of activations, MEFTs require slightly more training time than other PEFT methods and offer a slightly lower score when trained in FP16 instead of FP32. In the future, we are interested in applying MEFT to other areas, like computer vision and automatic speech recognition, and to other bigger backbones for more sequence-to-sequence tasks.

## Acknowledgements

We thank all reviewers for their great feedback. We also thank our colleague Yan Meng for her helpful review of our draft. This research was funded in part by the Netherlands Organization for Scientific Research (NWO) under project number VI.C.192.080.

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

# A    Related works

**Pre-trained language models**. PLMs, which are trained on extensive datasets for a common task such as predicting masked words [1, 2, 3, 26, 55, 56, 57, 58] or the next word [59, 60] in a sentence, play a vital role in facilitating knowledge transfer to downstream tasks. They have demonstrated remarkable achievements across various applications, consistently delivering state-of-the-art outcomes. Furthermore, scaling up PLMs has proven to yield predictable enhancements in performance for these downstream tasks [61, 62]. Consequently, the size of released PLMs has progressively grown, reaching an unprecedented scale of 100 billion parameters [9, 10, 11, 60, 63, 64]. Such large-scale PLMs unveil extraordinary capabilities, enabling zero-shot or in-context learning [59, 60] for a broad spectrum of tasks. Nevertheless, transfer learning remains a prevalent approach for effectively deploying these models in new task scenarios [29, 65, 66], which post unparalleled requirements on the computing resources.

**Parameter-efficient fine-tuning**. With the advent of large-scale PLMs, a new method that aims to reduce storage requirements, PEFT, has been proposed [14, 15, 19]. PEFT adds and trains a small number of parameters while matching the performance of full fine-tuning. There are various ways to add new parameters. For example, Houlsby et al. [14] and Pfeiffer et al. [16] insert small bottleneck modules (adapters) to the PLM. LoRA [17] injects rank decomposition matrices into the pre-trained weights. HiWi [13] inserts the pre-trained parameters to a low-rank adapter. (IA)$^3$ [29] scales the pre-trained weight with a trainable vector. Prompt-based methods [15, 19] append a sequence of trainable vectors to the word embeddings or attention components. Recently, some unified methods, which combine multiple PEFT methods in a heuristic way [18] or with the technique of neural architecture search [49, 67, 68], have also been proposed. Though PEFTs save the storage by a large margin compared to full fine-tuning, they still require a similar memory footprint during training as full fine-tuning [20, 21] because of the activation memory.

**Memory-efficient training**. Memory-efficient training aims to reduce the memory footprint during the training process. Reversible neural networks [40, 41, 42] reduce the activation memory by recomputing the activations with the outputs during back-propagation. Gradient checkpointing [69] trade computation for memory by dropping some intermediate activations and recovering them from an extra forward pass. The activation memory is $\mathcal{O}(1)$ and $\mathcal{O}(\sqrt{N})$ for reversible neural networks and gradient checkpointing, respectively. MEFT is the first method that is proposed to modify a PLM to its reversible variant. When applying MEFT on a deeper model, one can use gradient checkpointing to further reduce the activation memory for the layers with vanilla gradient.

Network compressions, like pruning [70, 71] and knowledge distillation [22, 23, 72], save the memory footprint for both training and inference. They compress a PLM to a smaller model by either deleting unimportant parameters or distilling knowledge from the PLM to the smaller model. Treating a PLM as a feature extractor and avoiding its gradient calculation is also an effective way to reduce the activation memory [20, 21]. However, these methods normally require extra pre-training before fine-tuning, or achieve a lower performance compared to full fine-tuning when using the same PLM.

# B    Limitations

We acknowledge the main limitation of this work is that we only evaluate our proposed methods on a limited amount of tasks and don't conduct experiments on the encoder-decoder models. The main reason for the limited amount of tasks is that our computing resources are constrained. In addition, the major criterion for our selection of the underlying models is that we could find many strong baselines on them without reproduction. BERT and RoBERTa fulfill this criterion very well on the GLUE benchmark. Regarding the encoder-decoder model, recently there is a clear trend of applying a decoder-only model on sequence-to-sequence tasks. Therefore, we apply OPT in this paper and plan to include LLAMA [11] for the instruction-finetuning data in the future.

Another limitation of MEFT is its lower score when trained in FP16 and on a deeper model. We have discussed this problem in §4.2. In sum, more reconstruction error is introduced by FP16 due to its numerical range and by a deeper model because of the error accumulation. Fortunately, the results are still comparable to the PEFT baselines when trained in FP16. Even trained in FP32, the activation memory footprints don't increase compared to FP16. One only needs to spend more training time in FP32 when using the same batch size as in FP16 (about 20% more training time). However, since

MEFTs reduce the memory footprint, a larger batch size during training is possible, which can save some training time. For deeper models, we offer a practical and effective setting in Figure 7.

Last but not least, when fine-tuning larger models, like $OPT_{1.3B}$ and $OPT_{6.7B}$ [9], the peak memory footprint is occupied by the model parameters rather than the activation (see Table 3). One needs to combine other techniques with MEFT to reduce the peak memory footprint, like loading the model in FP16 or even in int8 rather than in FP32, combining MEFT with ZeRO [73] as in Table 6.

## C    Step-by-step design for MEFT$_1$

For the reader's easy understanding, in this section, we explain MEFT$_1$ step-by-step. First, let's re-emphasize the guiding principles for our design: (1) For each reversible layer, we must have two inputs and two outputs as in Figure 3a. (2) We need to follow the starting point hypothesis. I.e. whenever we modify a PLM layer, we need to ensure the modified layer has almost the same output as the original PLM layer if we input the same input of the original PLM layer to the modified layer at the beginning of training. If the outputs are not similar, they become even more dissimilar after multiple layers, tearing down the PLM's initialization.

As shown in Figure 8a, for the first PLM layer, $h_0$ is the input and $h_1$ is the output. In Figure 8b, the inputs to the first reversible layer is $h_0^1 = h_0^2 = h_0$. Recapping the architecture of $\mathcal{F}_1$ in Figure 4c (up), we simply insert an adapter in parallel to the two consecutive feed-forward layers, and initialize the adapter as $W_{down}, W_{up} \sim \mathcal{N}(0, 0.02^2)$, which results in $h_1 \approx \mathcal{F}_1(h_0^2)$ since $h_0^2 = h_0$. If we set $\lambda \to 0$, $h_1^1 = \lambda h_0^1 + \mathcal{F}_1(h_0^2) \approx h_1$. In this way, $h_1^1$ plays the role of preserving the starting point. Now let's consider $h_1^2$. Due to our initialization of the adapter, the output from $\mathcal{G}_1$ ($\mathcal{G}_1$ is simply an adapter as in Figure 4c (down)) is close to $\mathbf{0}$. So $h_1^2 = \beta h_0^2 + \mathcal{G}_1(h_1^1) \approx \beta h_0 + \mathbf{0} = \beta h_0$. After switching the order of $h_1^1$ and $h_1^2$, $h_1^1 \approx \beta h_0$ and $h_1^2 \approx h_1$.

For the second reversible layer, if we don't switch the order of $h_1^1$ and $h_1^2$, it looks like Figure 8c. The input to $\mathcal{F}_2$ is $\beta h_0$, which breaks down the representation continuity of a PLM since the input to the pre-trained $\mathcal{F}_2$ should be close to $h_1$. If we switch their order as in Figure 8d, we preserve the representation continuity. And it results in $h_2^1 = \lambda \beta h_0 + \mathcal{F}_2(h_1) \approx h_2$ due to $\lambda \to 0$ and $h_2 \approx \mathcal{F}_2(h_1)$. Similar to the first reversible layer, $h_2^2 \approx \beta h_1$. After switching, $h_2^1 \approx \beta h_1$ and $h_2^2 \approx h_2$. By analogy, for the $n^{th}$ reversible layer, $h_n^1 \approx \beta h_{n-1}$ and $h_n^2 \approx h_n$.

After the final layer, we simply take the mean of two outputs as $h_N' = (h_N^1 + h_N^2)/2$, and input $h_N'$ to a task-specific head, like a classification layer. The design procedure is similar for MEFT$_2$ and MEFT$_3$. In sum, order switching is mainly for preserving the representation continuity, and setting the scaling factors close to 0 is mainly for preserving the starting point.

## D    Implementation details of the question-answering tasks

Compared to GLUE tasks where all tasks are classification tasks and the classification heads are randomly initialized, the question-answering tasks are sequence-to-sequence tasks and need the pre-trained output layer that shares the same parameters as the word embedding layer. The output

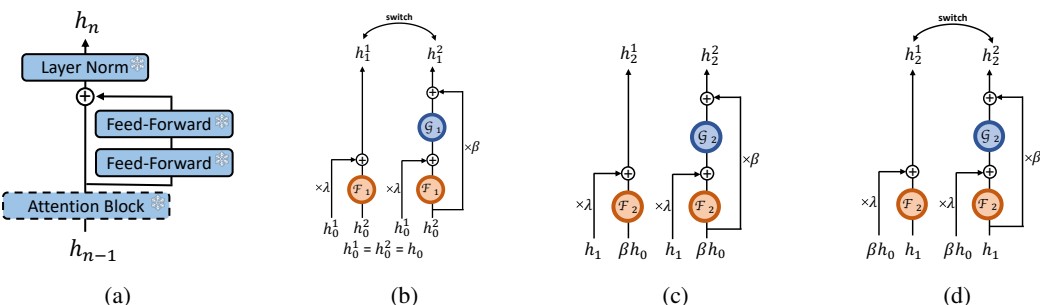

Figure 8: (a) The $n^{th}$ PLM layer; (b) The first MEFT$_1$ layer; (c) The second MEFT$_1$ layer without order switching; (d) The second MEFT$_1$ layer.

layer requires the continuity of representation. I.e. at the beginning of training, the input to the output layer, $h'_N$, should be close to $h_N$. Therefore, we need to do a modification to $h'_N$ instead of using $h'_N = (h^1_N + h^2_N)/2$.

Here we introduce a new scaling factor $\gamma$ and require $\gamma \to 0$. For MEFT$_1$, since $h^2_N \approx h_N$ (see Table 1), we set $h'_N = \gamma h^1_N + h^2_N \approx h^2_N \approx h_N$. Similarly, $h'_N = h^1_N + \gamma h^2_N \approx h^1_N \approx h_N$ for MEFT$_2$, and $h'_N = \gamma h^1_N + h^2_N \approx h^2_N \approx h_N$ for MEFT$_3$. Without any tuning, we set $\gamma = 0.1$ as other tuned scaling factors by default.

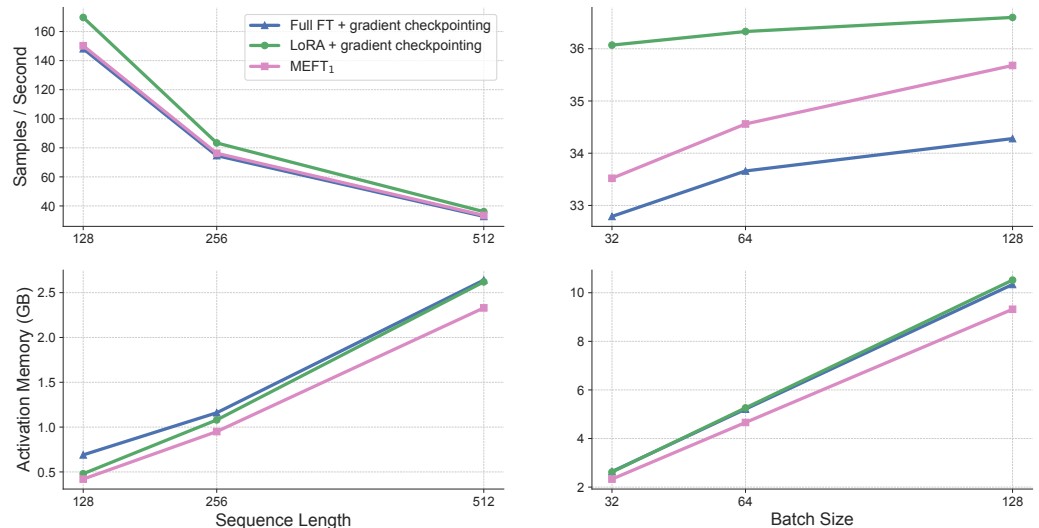

Figure 9: Throughput and activation memory for different sequence length and batch sizes on BERT$_\text{base}$. By default, the sequence length is 512 and the batch size is 32. For your reference, LoRA's throughput is 52.7 samples/second without gradient checkpointing for the default setting. Overall, MEFT shares the same level of throughput as LoRA with gradient checkpointing, while it is the lower bound of the activation memory for different settings.

# E    More results

## E.1    Compared to gradient checkpointing

Previously, we only theoretically stated that the activation memory for reversible network and gradient checkpointing is $\mathcal{O}(1)$ and $\mathcal{O}(\sqrt{N})$, respectively. In addition, we didn't compare the training time of MEFT with PEFT in detail. Here we offer some empirical results for your better understanding.

In Figure 9, we compare activation memory and throughput among MEFT , LoRA with gradient checkpointing and Full FT with gradient checkpointing. The throughput for all three methods is at the same level, maximum 12% difference between LoRA and MEFT when the sequence length is 128 and the batch size is 32. With an increasing sequence length, the gap becomes narrower to 7.5%. Notably, the throughput for LoRA without gradient checkpointing is 52.7 samples/second. With gradient checkpointing, it is 36.1 samples/second, 69% of the original throughput. For MEFT with the same setting, it is 33.5 samples/second, 64% of LoRA's throughput without gradient checkpointing. In sum, MEFT's throughput is at the same level as LoRA's with gradient checkpointing, and about 64% of LoRA's without gradient checkpointing. In addition, MEFT's activation memory is always the lower bound among these three methods. The gap between LoRA with gradient checkpointing and MEFT becomes larger with an increasing sequence length and batch size.

## E.2    Compared to quantization methods

Quantization is an orthogonal method to MEFT, which reduces the memory footprint of training or inference by reducing the parameter size to fewer bits and using low-bit-precision matrix multiplication. There are mainly three different quantization methods: (1) Post-training quantization

Table 5: Compared to QLoRA. $r = 8$ for all methods. Experimental setting stays the same as the default setting in Figure 9.

| Method | Activation Memory (GB) | Samples/Second |
|---|---|---|
| LoRA + gradient checkpointing | 2.62 | 36.1 |
| QLoRA + gradient checkpointing | 2.97 | 8.7 |
| $MEFT_1$ | 2.33 | 33.5 |

Table 6: Combine MEFT with ZeRO.

| Method | Peak Memory (GB) | Activation Memory (GB) |
|---|---|---|
| $MEFT_1$ | 28.2 | 8.2 |
| $MEFT_1$ + ZeRO | 6.4 | 6.4 |

[74, 75] that quantizes a trained model after pre-training or fine-tuning; (2) Lower-bit optimizer [76] that stores the optimizer state with lower precision and de-quantizes it only for the optimization, similarly to FP16 or BF16 mixed precision training but with lower-bit; (3) Lower-bit frozen LLM with LoRA, i.e. QLoRA [77], that applies 4-bit quantization to compress the LLM. During fine-tuning, QLoRA backpropagates gradients through the frozen 4-bit quantized LLM into the low-rank adapters. Notably, the computation data type for QLoRA is BF16. It de-quantizes weights to the computation data type to perform the forward and backward passes.

To some extent, all these three methods are orthogonal to our method and can be combined with MEFT: (1) Post-training quantization is mainly for reference and it can be applied to any trained models; (2) 8-bit Adam can also be applied to any models trained based on a gradient; (3) QLoRA is a combination of (1) and (2). For QLoRA, we conducted some experiments on $BERT_{base}$ with the default setting as Figure 9. As shown in Table 5, $METF_1$ saves the most activation memory while having a similar throughput as LoRA with gradient checkpointing. The reason for the larger activation memory of QLoRA than LoRA is that it has an additional de-quantization step, which also causes its smallest throughput.

## E.3 Combine MEFT with ZeRO

ZeRO [73] saves memory by partitioning the model's parameters and optimizer state among GPUs or between GPU and CPU. This method is orthogonal to MEFT, since MEFT saves memory from activations. We conduct some experiments on $OPT_{1.3B}$ by combining our method with DeepSpeed [78] ZeRO stage 3 that offloading model's parameters and the optimizer state to CPUs. As shown in Table 6, ZeRO significantly reduces the memory footprint from the model's parameters, therefore reducing MEFT's peak memory from 28.2GB to 6.4GB.

Table 7: Fine-tuning settings. Check §4.2 for the fine-tuning setting on BART.

| Hyper-parameter | GLUE | | Question-Answering |
|---|---|---|---|
| | RTE, MRPC, STS-B, CoLA | SST-2, QNLI, QQP, MNLI | |
| Learning Rate | $\{5,6,7,8\} \cdot 10^{-4}$ | $\{3,4,5\} \cdot 10^{-4}$ | $\{1,3,5,7\} \cdot 10^{-4}$ |
| Batch Size | $\{16,32\}$ | $\{16,32\}$ | $\{8,16,32\}$ |
| Max Epochs | $\{20,40\}$ | $\{10,20\}$ | $\{3,5,10\}$ |
| Weight Decay | 0.1 | 0.1 | 0.1 |
| Max Gradient Norm | 1 | 1 | 1 |
| Warmup Ratio | 0.06 | 0.06 | 0.06 |
| Learning Rate Decay | Linear | Linear | Linear |

Table 8: Statics of datasets

| Task | RTE | MRPC | STS-B | CoLA | SST-2 | QNLI | QQP | MNLI-m | MNLI-mm |
|------|-----|------|-------|------|-------|------|-----|--------|---------|
| **#Training** | 2.5k | 3.7k | 5.8k | 8.6k | 67.4k | 104.7k | 363.8k | 392.7k | |
| **#Development** | 0.3k | 0.4k | 1.5k | 1k | 0.9k | 5.5k | 40.4k | 9.8k | 9.8k |

| Task | OpenBookQA | PIQA | ARC-E | ARC-C | SciQ |
|------|-----------|------|-------|-------|------|
| **#Training** | 5.0k | 16.1k | 2.3k | 1.1k | 11.7k |
| **#Development** | 0.5k | 3.1k | 2.4k | 1.2k | 1k |

Table 9: Statics of models

| Model | #Parameter | #Layer | $d_{model}$ | Size in FP32 (GB) |
|-------|-----------|--------|-------------|-------------------|
| BERT$_{base}$ | 110M | 12 | 768 | 0.4 |
| BART$_{large}$ encoder | 205M | 12 | 1024 | 0.8 |
| RoBERTa$_{large}$ | 355M | 24 | 1024 | 1.4 |
| OPT$_{1.3B}$ | 1.3B | 24 | 2048 | 5.2 |
| OPT$_{6.7B}$ | 6.7B | 32 | 4096 | 25.6 |

```python
def backward_pass(self, y1, y2, dy1, dy2):
    with torch.enable_grad():
        y1.requires_grad = True
        # The intermediate activations of G are stored
        g_y1 = self.G(y1)
        # Obtain the gradient of y1
        g_y1.backward(dy2, retain_graph=True)

    with torch.no_grad():
        x2 = (y2 - g_y1) / self.x2_factor
        # Save memory, same for below
        del g_y1, y2
        dy1 += y1.grad
        # Save memory
        y1.grad = None

    with torch.enable_grad():
        x2.requires_grad = True
        # The intermediate activations of F are stored
        f_x2 = self.F(x2)
        # Obtain the gradient of x2
        f_x2.backward(dy1, retain_graph=False)

    with torch.no_grad():
        x1 = (y1 - f_x2) / self.x1_factor
        del f_x2, y1
        dy2 *= self.x2_factor
        # dy2=dx2, save memory by using the same variable
        dy2 += x2.grad
        x2.grad = None
        # dy1=dx1
        dy1 *= self.x1_factor
        x2 = x2.detach()
    return x1, x2, dy1, dy2
```

Listing 1: Backward pass for each Layer. The peak memory happens at Line 10 or Line 25, depending on whether the subnetwork $\mathcal{G}$ is larger than $\mathcal{F}$ or the opposite. In the code, we use x1, x2, y1, y2, x1_factor, x2_factor to represent $\boldsymbol{h}_{n-1}^1$, $\boldsymbol{h}_{n-1}^2$, $\boldsymbol{h}_n^1$, $\boldsymbol{h}_n^2$, $\lambda$ and $\beta$, respectively.

Table 10: Compared to $\mathcal{Y}$-Tuning on RoBERTa$_{\text{large}}$. We exclude the memory of $\mathcal{Y}$-Tuning for BART in Table 2, because it was not reported. Instead, the memory usage of $\mathcal{Y}$-Tuning for RoBERTa$_{\text{large}}$ was reported. Notably, the STS-B task is excluded from the calculation of the average score, because it was not evaluated in Liu et al. [20].

| Model | #Parameter | Peak Memory (GB) | Average Score |
|---|---|---|---|
| Full FT | 100% | 11.47 | **88.4** |
| LoRA | **0.23%** | 6.11 | 88.1 |
| $\mathcal{Y}$-Tuning | 4.57% | 2.08 | 82.1 |
| MEFT$_1$ | **0.23%** | 3.63 | **88.4** |

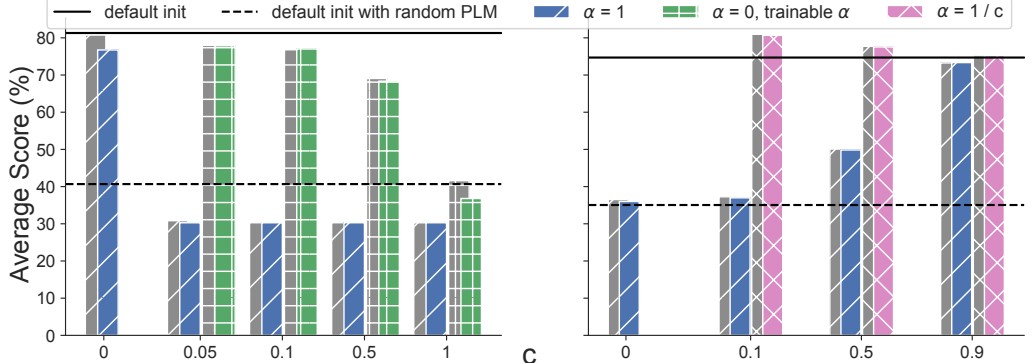

Figure 10: The initialization effect for PEFT, Left: LoRA, Right: (IA)$^3$. Instead of initializing $\boldsymbol{W}_{up} = \boldsymbol{c}$ like Figure 2b, here we initialize it as $\boldsymbol{W}_{up} \sim \mathcal{N}(c, 0.02^2)$, which should be more suitable for training due to its asymmetry. For convenient comparison, the results of $\boldsymbol{W}_{up} = \boldsymbol{c}$ (in grey) are also included. Overall, the results between $\boldsymbol{W}_{up} = \boldsymbol{c}$ and $\boldsymbol{W}_{up} \sim \mathcal{N}(c, 0.02^2)$ are comparable. However, when $c = 0$ for LoRA, the result of Gaussian initialization is slightly worse than the constant initialization. This further supports our starting point hypothesis, since the Gaussian initialization can't guarantee the output from the adapter is strictly equal to zero at the beginning of fine-tuning.

