# OpenReview forum: "Make Pre-trained Model Reversible: From Parameter to Memory Efficient Fine-Tuning"
_NeurIPS.cc/2023/Conference — NeurIPS 2023 poster_

### Official Review · Reviewer_X2di · 2023-06-13

**Soundness:** 3 good
**Presentation:** 3 good
**Contribution:** 3 good
**Rating:** 6
**Confidence:** 4

**Summary:**

>**Rebuttal:** The provided details satisfy my concerns. I think this paper should be accepted after applying the agreed changes.

>**TL;DR:** The proposed MEFT reduces training activation memory, which is an important topic for the ML community. MEFT outperforms existing PEFT techniques while consuming up to 84% less memory compared to full fine-tuning. However, MEFT comes with significant computation overhead, which is not explored. Furthermore, some key comparisons, such as gradient checkpointing, are missing. Addressing my concerns and questions would improve my score.

This paper proposes the Memory Efficient Fine-Tuning (MEFT) technique that achieves similar accuracy to that of traditional PEFT techniques while consuming up to 84% less memory compared to full fine-tuning. The paper discusses previous PEFT techniques and provides insights on key attributes for them to work. These insights include empirical results and detailed explanations. One of these key insights is the weights initialization. The paper shows the importance of initializing the PEFT weights such that they do not have any effect on the underlying LLM before starting the training process.

The paper proposes to combine reversible models with PEFT existing techniques to create MEFT algorithms that gain from both worlds. The paper provides three different variants of MEFT, which are explained as well as illustrated. The three variants of MEFT are empirically validated on four different architecture backbones BERT, RoBERTa, BART and OPT. The MEFT variants are compared to several popular PEFT techniques as well as some memory efficient techniques. These comparison are conducted on the GLUE dataset.

**Strengths:**

* **S.1.** The proposed MEFT algorithm tackles an important problem in existing PEFT techniques, which makes LLM PEFT training more accessible to researchers with low resources by reducing the activation memory up to 84% of full fine-tuning.
* **S.2.** The paper provides interesting insights that are explained and backed by results and detailed illustrations.
* **S.3.** The proposed MEFT variants outperform existing PEFT techniques on GLUE dataset with several architecture backbones.
* **S.4.** Reproduction code is provided as part of the submission.


**Weaknesses:**

* **W.1.** The training time of MEFT is not well explored and is only noted as "double the training time". Hence, the tradeoff between memory and computation efficiency is not well understood.
* **W.2.** Popular existing memory efficient training techniques such as tensor rematerialization (gradient checkpointing) [2][3] and ZeRO [1] are not compared to. These techniques sacrifice training speed for reduction in memory consumption, similar to the goal of MEFT. Furthermore, these techniques are compatible with existing PEFT approaches. Finally, these techniques are widely available in deep learning frameworks such as PyTorch, DeepSpeed, and Hugging Face Accelerate which makes comparison relatively simple.
* **W.3.** The font sizes in Figures 1,2,3,5, and 6 are too small which makes it very difficult to read. Furthermore, some figures are not self explanatory, such as Figure 5 which does not provide any explanation for the difference between the solid and dotted lines. These problems burden the reading flow of the paper.

**Typos.**
* Line #86: "for obtaining" → "to obtain"
* Line #86: "though we" → "even though we"
* Line #95: "in Figure 2a" → "in Figure 2a,"
* Line #97: "though with" → "even with"
* Line #106: "from an modified" → "from a modified"
* Line #114: "the added and" → "the added"


[1] Ren, J., Rajbhandari, S., Aminabadi, R.Y., Ruwase, O., Yang, S., Zhang, M., Li, D. and He, Y., 2021, July. ZeRO-Offload: Democratizing Billion-Scale Model Training. In USENIX Annual Technical Conference (pp. 551-564).

[2] Jain, P., Jain, A., Nrusimha, A., Gholami, A., Abbeel, P., Gonzalez, J., Keutzer, K. and Stoica, I., 2020. Checkmate: Breaking the memory wall with optimal tensor rematerialization. Proceedings of Machine Learning and Systems, 2, pp.497-511.

[3] Beaumont, O., Eyraud-Dubois, L. and Shilova, A., 2021. Efficient combination of rematerialization and offloading for training dnns. Advances in Neural Information Processing Systems, 34, pp.23844-23857.

**Questions:**

* **Q.1.** The paper proposes MEFT which sacrifices computation speed at the cost of lower peak memory. There are several existing common approaches (such as gradient checkpointing and DeepSpeed) for general memory efficient training which are compatible with PEFT techniques. Why are these comparisons not explored or detailed?
* **Q.2.** In the LoRA modified initialization method, both $W_{up}$ and $W_{down}$ are initialized with a single constant. This causes unwanted symmetries, which makes it more difficult to learn. An alternative approach would be to initialize $W_{down}=1$ (as tested in this paper, but instead of initializing $W_{up}$ to a constant, initialize it with a random gaussian distribution where $c$ can act as the distribution mean. Does this make sense?
* **Q.3.** In Line #175 it is noted that the adapter weights' $W_{down}$ and $W_{up}$ are initialized from a random gaussian distribution with zero mean. It is further noted, that the outputs of the adapter would be close to zero. However, in practice this does not hold true,  and the actual outputs can significantly deviate from the desired zero output. What am I missing here?
* **Q.4.** In Figure 6 it seems that LoRA only reduces slightly the peak memory in comparison to the full fine-tuning. However, the optimizer states, which are generally 3 times the size of the model, are significantly smaller in LoRA. Does this result hold for other LLMs and how does the batch size affect it as well?
* **Q.5.** The MEFT technique does not seem to be only compatible with LLMs and can be compatible with other domains, as sted in the conclusion section. However, the experimental results are solely on LLMs. What is the reason for that?
* **Q.6.** In Table 2 the memory usage of Y-Tuning is missing, why is that?


**Limitations:**

The limitations are discussed in paragraph line #315. These limitations include limited compatibility with quantization techniques due to training instabilities and significantly longer training time due to activation re-computation.

---

> ### Author Rebuttal · Authors · 2023-08-10
>
> Dear reviewer, we would like to thank you for your time, effort, and thorough reviews! We are really encouraged by the reviews that highlight our work:
>
> (1) Tackles an important problem in the existing PEFT methods.
>
> (2) Offers insightful observation for the success of existing PEFT methods.
>
> (3) Offers competitive results on different pre-trained models.
>
> We have responded to your comments below and updated our draft accordingly by improving the presentation, citing missing references, and adding new baselines. If you still have additional questions and suggestions, we would be happy to answer your questions and incorporate the suggestions into an updated draft.
>
> > W.1 and Limitations. The computation overhead/ training time of MEFT is not well explored.
>
> Thank you for this advice. We have conducted related experiments and show the results in the $1^{th}$ point of the general response.
>
> > W.2 and Q.1. Missing baselines from gradient checkpointing and ZeRO.
>
> Thank you for this advice. We have conducted related experiments and show the results in the $1^{st}$ and $3^{rd}$ point of the general response.
>
> > W.3. Too small font size and less self-explanatory captions for the figures.
>
> Thanks for this advice. We already made a change to the font size and caption in the uploaded PDF for new results. All other figures have been adjusted in our updated draft.
>
> > Q.2. Is it better to initialize $W_{up}$ ~ $\mathcal{N}(c, \sigma^2)$ instead of $W_{up} = c$?
>
> This is a very inspiring observation. According to your suggestion, we repeat the experiments in Figure 2 by initializing $W_{up}$ ~ $\mathcal{N}(c, 0.02^2)$, and show the results in Figure 10 in the uploaded PDF in the general response. **Overall, their results are similar**. One reason might be the limited amount of updated parameters, only about 0.27%, which makes the optimization easy.
> Another interesting observation is the slight degradtion of $W_{up}$ ~ $\mathcal{N}(0, 0.02^2)$ when comparing to $W_{up} = 0$ for LoRA. We argue the main reason is: $W_{up}$ ~ $\mathcal{N}(0, 0.02^2)$ can't guarantee that the output from the adapter is strictly equal to zero, which further supports our starting point hypothesis.
>
> > Q.3. In practice, the actual output may significantly deviate from the desired zero output.
>
> This is a very good observation. We conduct two experiments to check it.
>
> (1) We measure the output from $\mathcal{G}$ in Figure 4 (a) on the development set of RTE. For most samples, it is very close to 0. We argue the main reason is both $W_{down}$ and $W_{up} \sim \mathcal{N}(0, 0.02^2)$. The multiplication of them, $W_{up}GeLU(W_{down}h)$ makes the output very small.
>
> (2) We initialize $W_{up} = 0$ as LoRA to strictly make this output equal to 0. The result is very similar to our answer to your Q.2. We obtain a slightly better score, about +0.2. We will add this new result to our updated version.
>
> > Q.4.Limited reduction of peak memory from LoRA in Figure 6.
>
> Sorry for the not-well self-explanatory captions and the y label. As our statement in Line 295, the memory shown in Figure 6 is the activation memory instead of the peak memory. Since LoRA still caches most of the activation memory as full fine-tuning, their activation memory is very similar.
>
> Regarding the peak memory, we list some related numbers of BERT$_{base}$ from Table 2 here:
>
> | Method | Peak Memory (GB) | Activation Memory (GB) |
> | -------- | -------- | -------- |
> | Full FT     | 16.67  | 14.98 |
> | LoRA | 13.45 | 13.02 |
>
> So the memory difference for the optimizer state between Full FT and LoRA is (16.67 - 14.98) - (13.45 - 13.02) = 1.26GB. It is about three times of the model's size of 0.45GB. Notably, we make an approximation here by assuming the memory footprint for LoRA's optimizer state is 0, since only 0.27% of parameters are trainable.
>
> The claim, the memory for the optimizer state is generally 3 times of the model's, holds for other LLMs if you train LLM with a second-order optimizer, like Adam, because one needs to save the first, second momentum, and also the calculated gradient. In addition, this memory for the optimizer state is invariant to the batch size. It's only related to the size of trainable parameters.
>
> > Q.5. Apply MEFT to other domains.
>
> Limited computation resource is the main reason for our limited application in the NLP area. Thanks to your advice, we transfer MEFT$_1$ to a CV task and show the result in the $4^{th}$ point of the general response.
>
> > Q.6. In Table 2 the memory usage of Y-Tuning is missing, why is that?
>
> The main reason for our exclusion of the Y-Tuning memory usage is that the original paper [7] doesn't show the memory usage for BART and also doesn't release their code. This offers us some difficulties to reproduce their results. But the memory usage for Y-Tuning should be quite low since it uses the pre-trained model as a feature extractor, and there is not any gradient calculation on the backbone model.
>
> For your reference, the peak memory usage for RoBERTa$_{large}$ is reported in [7]. It is 18.1% of full fine-tuning (Table 3 in [7]). The related results are:
>
> | Method | #Params | Peak Memory (GB) | Avg. |
> | ------ | ---------------- | --- | ---  |
> | Full FT| 100% | 11.47 | **88.4** |
> | LoRA | **0.23%** | 6.11 | 88.1 |
> | $\mathcal{Y}$-Tuning | 4.57% | **2.08** | 82.1 |
> | MEFT$_1$ | **0.23%** | 3.63 | **88.4** |
>
> Notably, STS-B task is excluded for this table, because it is not evaluated in [7]. We have included this table in our updated draft.
>
> > Limitation: Unstable training.
>
> About the unstable training, we want to make a clarification here. The unstable training with FP16 causes slight performance degradation when compared to training with FP32, but it doesn't cause failed training and is still comparable to other baselines.
>
> [7] Liu, Y., An, C., & Qiu, X. (2022). $\mathcal{Y}$-Tuning: An Efficient Tuning Paradigm for Large-Scale Pre-Trained Models via Label Representation Learning.

---

> > ### Comment · Reviewer_X2di · 2023-08-15
> > **Response to Rebuttal**
> >
> > Thank you for the detailed answers and results.
> >
> > The provided results and details satisfy my concerns. I will update my review accordingly.

---

> > > ### Author Response · Authors · 2023-08-16
> > > **Thank you for your advice to make our paper better**
> > >
> > > Thank you for your increased score!
> > >
> > > We really appreciate your discussion and advice. By including these new results, we believe the paper is strengthened than the previous version. If you still have further questions or guidance, we will gladly answer them and include them in our updated version.

---

### Official Review · Reviewer_dsz7 · 2023-06-25

**Soundness:** 2 fair
**Presentation:** 2 fair
**Contribution:** 2 fair
**Rating:** 4
**Confidence:** 4

**Summary:**

This paper points out that the existing parameter-efficient fine-tuning (PEFT) methods are not memory-efficient, since they need to cache most of the intermediate activations for the gradient calculation. To resolve this problem, the authors introduce to use the reversible model to reduce the activation memory. Specifically, this paper have these contributions: 1) it is important to preserve the weight of PLM at the initialization; 2) insert adapters and make it reversible without additional pre-training. Extensive results also demonstrate the effectiveness of the proposed method.

**Strengths:**

1. This paper points out the importance of memory-efficient fine-tuning in current pre-trained models. And the authors introduce the idea of RevNet into the pre-trained language model, and design three reversible variants to PLM.

**Weaknesses:**

1. Some conclusions seem naive. For example, preserving the starting point from the pre-trained model at the beginning of training is a common experience. Previous works like LoRA also point out the importance of this point, and this point is also a common method to avoid catastrophic forgetting.
2. Memory-efficient training is important, but the authors do not mention quantization in this paper.  Quantization has been considered as the one of most effective methods in LLM to save memory. While for RevNet or the proposed architecture, these reversible-based neural networks actually are a trade-off between computation and memory (it will bring more computation to save memory). So, I do not think the proposed method is advantageous to Quantization and directly comparing the proposed methods with PEFT may also not be appropriate.
3. Essentially, the proposed method adopts an adapter structure and then uses the reversible structure into the transformer and presents three patterns (which formulate different $\cal{F}$ and $\cal{G}$) and trains each network from the start point. The idea is to combine existing work (RevNet) with current PEFT methods, and then combine the initialization that preserving the starting point from the pretrained model. So, I think the contributions are limited.
4. In experimental results, authors report MEFT with vanilla gradient. I think such reports are confusion, since the main contribution in this paper is the reversible gradient with MEFT structure.

**Questions:**

1. This paper presents three variants of MEFT, but is there any conclusion that which one should be chosen under different scenarios?

**Limitations:**

The authors have presented limitations of their proposed works, and currently, these limitations are still a problem for the proposed methods. Besides, I also give some comments about the limitations of this paper (please see my comments in **Weaknesses**). I do not see any potential negative societal impact.

---

> ### Author Rebuttal · Authors · 2023-08-10
>
> Dear reviewer, we would like to thank you for your time, effort, and thorough reviews!
>
> We have responded to your comments below and updated our draft accordingly. If you still have additional questions and suggestions, we would be happy to answer your questions and incorporate the suggestions into an updated draft.
>
> > W.1. Some conclusions seem naive. Previous work already points out the importance of preserving the starting point.
>
> We acknowledge that preserving the starting point from the pre-trained model is not a new concept (we also claimed this in Lines 109-110). In this paper, we explore its significance by conducting thorough and well-established experiments. To the best of our knowledge, our paper is the first work to explore this concept. In addition, **we also have some non-intuitive findings from Figure 2(b):**
>
> **(1) When $c >= 0.05$, the results are even worse than the one with a randomly initialized model for LoRA.**
>
> **(2) A $\alpha$ with an initialization of 0 for LoRA or the inverse of $c$ for (IA)$^3$ can make the bad results become way better, even better than the default initialization for (IA)$^3$.**
>
> We understand that some conclusions might appear straightforward or "naive" when viewed in isolation. However, in the broader context of our research, these conclusions serve a pivotal role in building upon and connecting the various aspects of our study. We will ensure to emphasize the distinctions and novelties of our approach more clearly in the revised draft.
>
> > W.2. Missing comparison with quantization methods.
>
> Thank you for this advice. We have conducted new experiments on QLoRA [3] and show the discussion and results in the $2^{nd}$ point of the general response. Apart from quantization, we also combine our method with ZeRO [4] and show the results in the $3^{rd}$ point of the general response.
>
> >  W.2: The proposed method is not advantageous to quantization and directly comparing the proposed methods with PEFT may also not be appropriate.
>
> We politely disagree with this point. Quantization is also a trade-off between time/computation and memory. Taking QLoRA as an example (check the new results in the $2^{nd}$ point of general response), the throughput is 8.7 samples/second, while MEFT is 33.5 samples/second. Though both methods target the same issue, memory-constrained training, they focus on different aspects. Quantization reduces the memory footprint occupied by the model's parameters or the optimizer state, while MEFT reduces the activation memory.
>
> Regarding the inappropriate direct comparison with other PEFT methods, our intention is to show that the MEFT's task performance is comparable to PEFT's, while having the memory-efficient characteristics at the same time. In addition, we also included other strong memory-efficient baselines in Table 2, like Y-Tuning and LST. Thanks to your advice, we further include more baselines, like gradient checkpointing, quantization and ZeRO.
>
> > W.3: Limited contribution because of the combination of PEFT, RevNet and the starting point hypothesis.
>
> We acknowledge that our approach builds upon established techniques, but we'd like to highlight the novelty and advantages of our approach:
>
> (1) RevNet and Transformer share two distinct architectures, as shown in Figure 4. Converting a pre-trained non-reversible model to a reversible model poses a totally new challenge that is not well-explored.
>
> (2) By thoroughly investigating the key factor for the success of PEFT, we propose an intuitive hypothesis, the starting point hypothesis, which is the foundation of MEFT.
>
> (3) Through dedicated design, we convert a pre-trained Transformer to its reversible variant without sacrificing the performance, while having the advantage of saving activation memory.
>
> (4) As a side product, MEFT with vanilla gradient outperforms all baselines, serving as an effective PEFT method.
>
> We don't simply combine the techniques of PEFT and RevNet to obtain our final result. Instead, we offer a new way to explore PEFT and pre-trained models.
>
> > W.4: Confused results from MEFT with vanilla gradient.
>
> Sorry for showing MEFT with vanilla gradient and reversible gradient in the same table, which may distract your attention. We will split them in our updated version. But the results for MEFT with vanilla gradient are also important since they are the best among all baselines. It shows MEFT with vanilla gradient can serve as an effective PEFT method.
>
> > Q.1: Which variant of MEFTs should be chosen under different scenarios?
>
> According to Table 2, MEFT$_3$ should be chosen if one only wants to use MEFT as a PEFT method with a vanilla gradient. MEFT$_1$ should be chosen when concerned about the training memory.
>
> > Limitations: The stated limitations in the paper are still problems for MEFT.
>
> The main limitations we stated in this paper (Appendix B) are:
>
> (1) Limited amount of downstream tasks, only in NLP area with 8 classification tasks and 5 question-answering tasks.
>
> (2) Lower score when trained with FP16 and on a deeper model.
>
> (3) The memory footprint occupied by the model's parameters is larger than the one occupied by the activation for a large language model.
>
> For all three limitations, we have resolved them and offered practical solutions:
>
> (1) We successfully transfer our method to the computer vision area. Please read the $4^{th}$ point in the general response.
>
> (2) For the lower score when trained with FP16, the score is worse than the one trained with FP32. However, it is still comparable to other memory-efficient baselines, 81.9 vs 82.1 of LST. LST was also trained with FP32 and requires > 3 times activation memory. For the deeper model, we offered some practical settings in Figure 7 and showed its effectiveness in Table 2 and 3.
>
> (3) We can combine MEFT with ZeRO, reducing the peak memory from 28.2GB to 6.4GB.
>
> [6] Kitaev, N., Kaiser, Ł., & Levskaya, A. (2020). Reformer: The efficient transformer.

---

> > ### Comment · Reviewer_dsz7 · 2023-08-18
> >
> > Thanks for the response from the authors. However, I think this paper still has some limitations:
> >
> > 1. Authors argue that they proposed an intuitive hypothesis (i.e., the starting point hypothesis). However, just as said, this hypothesis is not supervised since it is a common experience and has been used in many techniques (e.g., LoRa, Zero-init Attention). Therefore, the contribution of this point is not enough. And authors argue they provide some non-intuitive findings. However, these findings are only based on empirical experiments, rather than theoretical analysis. Therefore, the conclusion of its generality still needs to be validated.
> > 2. Thanks for the authors' clarifications about the differences between quantization and MEFT. However, why I point out this problem is because the title called "From Parameter-Efficient to Memory-Efficient Fine-Tuning". But actually, the contribution of this paper is more like memory-efficient fine-tuning for PEFT. Don't you think such a title is over-claim? If that, some other work like FlashAttention-1/2, which is optimized from the perspective of the system and efficiently saves memory, and will not double the training time and affect any performance, should also be considered. Therefore, I admit that incorporating the thinking of RevNet into the pre-trained model is ok but MEFT is a very large topic. I suggest the authors check it again.

---

> > > ### Author Response · Authors · 2023-08-20
> > >
> > > > Authors argue that they proposed an intuitive hypothesis (i.e., the starting point hypothesis). However, just as said, this hypothesis is not supervised since it is a common experience and has been used in many techniques (e.g., LoRa, Zero-init Attention). Therefore, the contribution of this point is not enough.
> > >
> > > Here we make a summarization of some well-known PEFT methods for their default initialization:
> > >
> > > | Method | Initialization of added Parameters | Apply the starting point hypothesis? |
> > > | --- | --- | ---|
> > > | Adapter [8] | $\mathcal{N}(0, 0.02^2)$ | No |
> > > | Pfeiffer Adapter [9] | $\mathcal{N}(0, 0.02^2)$ | No |
> > > | LoRA | $W_{up} = 0$ | Yes |
> > > | (IA)$^3$ | $l = 1$ | Yes |
> > > | Prefix Tuning [10] | $\mathcal{U}(-1, 1)$ | No |
> > > | Zero-init Attention [11] (Released 10 days after the submission deadline) | zero gating | Yes |
> > >
> > > **To the best of our knowledge, there are only two works that implicitly apply the starting point hypothesis before the submission deadline. Our work is the first paper to investigate this hypothesis with some sophisticated designs.** Some previous works [8, 9, 11] use the default initialization from PyTorch.
> > >
> > > > And authors argue they provide some non-intuitive findings. However, these findings are only based on empirical experiments, rather than theoretical analysis. Therefore, the conclusion of its generality still needs to be validated.
> > >
> > > LoRA and (IA)$^3$ are the only two methods that implicitly apply the starting point hypothesis before our submission. We investigate the starting point hypothesis with these two methods on four different tasks. In Figure 2(b) and Figure 10 of the newly uploaded PDF, we conduct 192 experiments (3 random seeds with the mentioned grid search space on four tasks) for each bar, **total 3072 experiments conducted in Figure 2(b)**. Therefore, we believe the concluded findings are well-claimed, though they are only empirically tested.
> > >
> > > Unfortunately, we didn't provide the theoretical justification for our findings. We are open to any suggestions.
> > >
> > > > Thanks for the authors' clarifications about the differences between quantization and MEFT. However, why I point out this problem is because the title called "From Parameter-Efficient to Memory-Efficient Fine-Tuning". But actually, the contribution of this paper is more like memory-efficient fine-tuning for PEFT. Don't you think such a title is over-claim? If that is, some other work like FlashAttention-1/2, which is optimized from the perspective of the system and efficiently saves memory, and will not double the training time and affect any performance, should also be considered. Therefore, I admit that incorporating the thinking of RevNet into the pre-trained model is ok but MEFT is a very large topic. I suggest the authors check it again.
> > >
> > > We appreciate the reviewer's insight regarding the title of our work. Our intent was to convey the transition or enhancement from Parameter-Efficient Fine-Tuning (PEFT) to a method that's also memory-efficient. We understand that the title might give the impression that our work encompasses the entirety of MEFT. Based on this feedback, we are open to reconsidering the title to more accurately reflect the paper's contributions, such as "Memory-Efficient Fine-Tuning Enhancements for PEFT".
> > >
> > > Regarding FlashAttention-1/2, it exchanges the computation between SRAM and HBM to speed up the training and save memory. Like quantization, this method is orthogonal to our work, because it mainly solves the issue of the large memory raised by long sequences from the attention module, while our work focuses on reducing the activation memory. Our work can be combined with it without any modification on Transformers.

---

### Official Review · Reviewer_mzDQ · 2023-06-29

**Soundness:** 3 good
**Presentation:** 2 fair
**Contribution:** 3 good
**Rating:** 6
**Confidence:** 4

**Summary:**

Authors propose a memory-efficient (MEFT) variant of parameter-efficient fine-tuning (PEFT), which utilizes reversible networks to save memory stored for activations during model forward propagation. The authors showed insights into the reasons for the effectiveness of PEFT. They evaluated several variants of MEFT with GLUE and QA datasets, indicating that the proposed method does not produce performance gaps while reducing memory usage.

**Strengths:**

- The proposed method shows competitive results with benchmarks
- Experiments with the initialization of LoRa are insightful
- Generally, the proposed methods are useful for practitioners

**Weaknesses:**

- The presentation could be improved. E.g., I struggled with Section 2.3 since $h^1, h^2 $ (L136) was not defined above.
- It would be fascinating to compare the proposed method with gradient checkpointing used for training with LoRA/etc. Naively, the usage of invertible layers should be faster. However, seeing such an evaluation in the paper would be highly beneficial.

**Questions:**

Please refer to the weaknesses section.

**Limitations:**

–

---

> ### Author Rebuttal · Authors · 2023-08-09
>
> Dear reviewer, we would like to thank you for your time, effort, and thorough reviews! We are really encouraged by the reviews that highlight our work:
>
> (1) Offers competitive results;
>
> (2) Conducts insightful experiments for LoRA's initialization;
>
> (3) Is useful for practitioners.
>
> We have responded to your comments below and updated our draft accordingly. If you still have additional questions and suggestions, we would be happy to answer your questions and incorporate the suggestions into an updated draft.
>  > 1. The presentation could be improved. E.g., I struggled with Section 2.3 since  $h_n^1$ and $h_n^2$ (L136) was not defined above.
>
> Thank you for this advice. Compared to the previous annotation $h_n$ (Line 81) representing the input to the $n^{th}$ layer, we add a superscript to distinguish the two inputs to the reversible layer. We have added a clarification to it for our new draft.
>
> > 2. It would be fascinating to compare the proposed method with gradient checkpointing used for training with LoRA/etc. Naively, the usage of invertible layers should be faster. However, seeing such an evaluation in the paper would be highly beneficial.
>
> Thank you for this advice. We offer these new results in the 1$^{st}$ point of the general response. **In sum, MEFT shares the same level of training throughput as LoRA with gradient checkpointing, while it saves the most activation memory for different settings.**
>
> In addition, there is a misunderstanding in your statement. The throughput of MEFT and LoRA with gradient checkpointing is at the same level because both of them compute two forward passes and one backward pass. The second forward pass is for the activation reconstruction.

---

> > ### Author Response · Authors · 2023-08-22
> > **Thank you for the increased score**
> >
> > We are really encouraged by the increased score. And thank you for the advice on incorporating the comparison to gradient checkpointing, which makes our paper more solid.

---

### Author Rebuttal · Authors · 2023-08-10

We would like to thank all reviewers for your time, effort, and thorough reviews! Regarding typos, presentation improvement, and missing baselines, we have incorporated suggestions into our updated draft.

Here we offer new results for some common concerns:

> 1. Compare MEFT to gradient checkpointing.

In our previous submitted materials (Line 728 in Appendix A), we only theoretically stated that the activation memory for reversible network and gradient checkpointing is $\mathcal{O}(1)$ and $\mathcal{O}(N)$, respectively. In addition, in Line 75, we also only broadly stated that the training time of MEFT is about double the training time of PEFT. Both are not well-explored and may not be intuitive enough. Here we offer some experimental results for these two claims.

In Figure 9 of the uploaded PDF, we compare activation memory and throughput among MEFT$_1$, LoRA with gradient checkpointing and Full FT with gradient checkpointing. The throughput for all three methods is at the same level, maximum 12% difference between LoRA and MEFT$_1$ when the sequence length is 128 and the batch size is 32. With an increasing sequence length, the gap becomes narrower to 7.5%. Notably, the throughput for LoRA without gradient checkpointing is 52.7 samples/second. With gradient checkpointing, it is 36.1 samples/second, 69% of the original throughput. For MEFT$_1$ with the same setting, it is 33.5 samples/second, 64% of LoRA's throughput without gradient checkpointing. **In sum, MEFT's throughput is at the same level as LoRA's with gradient checkpointing, and about 64% of LoRA's without gradient checkpointing**, which is better than our previously reported safe number, i.e. 50%.
In addition, **MEFT's activation memory is always the lower bound among these three methods**. And the gap between LoRA with gradient checkpointing and MEFT becomes larger with an increasing sequence length and batch size.

> 2. Compare MEFT to quantization methods.

There are mainly three different quantization methods:

(1) Post-training quantization [1]: Quantize a trained model to a lower bit after pre-training/fine-tuning;

(2) Lower-bit optimizer [2]: Similar to the idea of mixed precision training, it stores the optimizer state with lower precision and dequantizes it only for the optimization;

(3) Lower-bit frozen LLM with LoRA, i.e. QLoRA [3] (released one week after the submission deadline): QLoRA applies 4-bit quantization to compress the LLM. During fine-tuning, QLoRA backpropagates gradients through the frozen 4-bit quantized LLM into the Low-Rank Adapters. Notably, the computation data type for QLoRA is BF16. It dequantizes weights to the computation data type to perform the forward and backward passes.

We summarize the main benefits of these three methods here:

| Save Memory from | Model's Parameter | Optimizer State | Activation |
| -------- | -------- | -------- | -----|
| post-training quantization [1] | Yes     | No     | No |
| 8-bit Adam [2] | No | Yes | No |
| QLoRA [3] | Yes | Yes | No |

Note: QLoRA also introduces paged optimizers to reduce the optimizer states, which is similar to DeepSpeed ZeRO stage 1 where the optimizer state is partitioned.

To some extent, all these three methods are orthogonal to our method and can be combined with MEFT:

(1) Post-training quantization is mainly for reference and it can be applied to any trained models.

(2) 8-bit Adam can also be applied to any models trained based on gradient.

(3) QLoRA is a combination of (1) and (2).

For QLoRA, we conduct some experiments on BERT$_{base}$ with the default setting as Figure 9:

| Method   | Activation Memory (GB) | Samples / second |
| -------- | -------- |   ----- |
| LoRA + gradient checkpointing  | 2.62 | 36.1 |
| QLoRA + gradient checkpointing | 2.97 | 8.7 |
| MEFT$_1$ |  2.33 | 33.5 |

**METF$_1$ saves the most activation memory while having a similar throughput as LoRA with gradient checkpointing.** The reason for the larger activation memory of QLoRA than LoRA is that it has an additional dequanzation step. Because of this, its throughput is the smallest.

> 3. Combine MEFT with ZeRO [4].

ZeRO [4] saves memory by partitioning the model's parameters and optimizer state among GPUs or between GPU and CPU. This method is orthogonal to our method. We conduct some experiments on OPT1.3B by combining our method with DeepSpeed ZeRO stage 3 that offloading model's parameters and the optimizer state to CPUs:

| Method   | MEFT$_1$ | MEFT$_1$ + ZeRO (offloading to CPU) |
| -------- | -------- | -------- |
| Peak / Activation Memory (GB)  | 28.2 / 8.2 | 6.4 / 6.4

ZeRO significantly reduces the memory footprint for model's parameters, therefore reducing MEFT's peak memory from 28.2GB to 6.4GB.

> 4. Result on CV task.

By simply borrowing the default training settings from AdaptFormer [5], we obtain the following results on the SVHN task. More CV tasks are on our TODO list. Due to limited computation resources, we only obtain the result on SVHN for now.

| Method | Acc@1 | Peak memory (GB) |
| --- | --- | ---|
| Full FT | 97.67 |  - |
| AdaptFormer | 96.89 | 36 |
| MEFT$_1$ | 96.74 | 9 |

**Similar to NLP's results, MEFT achieves comparable performance as AdaptFormer while saving a large amount of memory footprint.**

[1] Frantar, E., Ashkboos, S., Hoefler, T., & Alistarh, D. (2022). Gptq: Accurate post-training quantization for generative pre-trained transformers.

[2] Dettmers, T., Lewis, M., Shleifer, S., & Zettlemoyer, L. (2021). 8-bit optimizers via block-wise quantization.

[3] Dettmers, T., Pagnoni, A., Holtzman, A., & Zettlemoyer, L. (2023). Qlora: Efficient finetuning of quantized llms.

[4] Ren, J., Rajbhandari, S., Aminabadi, R.Y., Ruwase, O., Yang, S., Zhang, M., Li, D. and He, Y., 2021, July. ZeRO-Offload: Democratizing Billion-Scale Model Training.

[5] Chen, S., Ge, C., Tong, Z., Wang, J., Song, Y., Wang, J., & Luo, P. (2022). Adaptformer: Adapting vision transformers for scalable visual recognition.

---

### Decision · Program_Chairs · 2023-09-21

**Decision:**

Accept (poster)

**Comment:**

The paper proposes a method for memory-efficient fine-tuning and recent parameter-efficient fine-tuning (PEFT) still require caching the intermediate activations for the gradient calculation. This topic is of practical importance. All reviewers agreed that the proposed method is novel, borrowing ideas from RevNet and applying them to Transformers. The paper also provides insights regarding the importance of PEFT initialisation. All reviewers agreed that the experiments and results were solid. In addition, the authors provided a very detailed rebuttal. Incorporating the additional results and clarifications in the paper will strengthen it and I would encourage the authors to do so.